# Polymeric Micellar Systems—A Special Emphasis on “Smart” Drug Delivery

**DOI:** 10.3390/pharmaceutics15030976

**Published:** 2023-03-17

**Authors:** Irina Negut, Bogdan Bita

**Affiliations:** 1National Institute for Laser, Plasma and Radiation Physics, 409 Atomistilor Street, Magurele, 077125 Bucharest, Romania; 2Faculty of Physics, University of Bucharest, 077125 Măgurele, Romania

**Keywords:** polymeric micelles, anti-cancer drugs, drug delivery, stimuli-responsive micelles

## Abstract

Concurrent developments in anticancer nanotechnological treatments have been observed as the burden of cancer increases every year. The 21st century has seen a transformation in the study of medicine thanks to the advancement in the field of material science and nanomedicine. Improved drug delivery systems with proven efficacy and fewer side effects have been made possible. Nanoformulations with varied functions are being created using lipids, polymers, and inorganic and peptide-based nanomedicines. Therefore, thorough knowledge of these intelligent nanomedicines is crucial for developing very promising drug delivery systems. Polymeric micelles are often simple to make and have high solubilization characteristics; as a result, they seem to be a promising alternative to other nanosystems. Even though recent studies have provided an overview of polymeric micelles, here we included a discussion on the “intelligent” drug delivery from these systems. We also summarized the state-of-the-art and the most recent developments of polymeric micellar systems with respect to cancer treatments. Additionally, we gave significant attention to the clinical translation potential of polymeric micellar systems in the treatment of various cancers.

## 1. Introduction

With a high frequency of breast (~2.26 million), lung (~2.21 million), colorectal (~1.9 million), and prostate (~1.4 million) cases, cancer is the second major cause of death in the world [1]. Although there has been much improvement in treating cancer, a full recovery remains a pipe dream. There are several anticancer medications that are available as monotherapies and combination therapies to slow the spread of cancer. Nevertheless, most of the drugs that are used clinically belong to biopharmaceutical classification system classes (BCS) III or IV owing to their poor solubility, dissolution, permeability, and bioavailability. These drugs include paclitaxel (PTX), docetaxel, cisplatin, methotrexate, etoposide, and bleomycin [2]. Their low aqueous solubility and poor permeability make them unable to reach the tumor site at the desired therapeutic concentration. This problem necessitates the use of high doses of potent anticancer drugs and multidrug therapies, leading to unwanted toxicities that result from the non-specific distribution of drugs throughout the body. Furthermore, multidrug resistance and the lack of early diagnostic approaches are also major challenges associated with the treatment of cancer [3].

The insufficient in vivo stability, immunogenicity, non-specific targeting, negative charge, large size, and hydrophilicity of the small molecules (such as genes, siRNAs (small interfering RNA), and plasmids) used to treat various types of cancer limit their curative efficacy in preventing cancer progression [2,3].

Nanosystems made of polymers are effective platforms for myriad targeted treatments. Numerous tumor-targeting nanoparticles have been developed for use in cancer applications as a result of the growing interest in applying nanotechnology to cancer detection and treatment [4,5]. In order to address a synergistic strategy to combat metastatic tumors, nanomedicines are currently being used to entrap traditional therapeutics and diagnostic equipment [6,7,8]. According to the current circumstances, developing new delivery systems that are quick and simple to prepare, affordable, have high drug loading efficiency, are stable under biological conditions, encourage drug retention in tissues, and are tailored for the delivery of already-approved FDA-approved drugs is the quickest and most practical way to improve the cancer treatment regimen [9]. In this regard, polymeric micellar systems have demonstrated significant promise for the administration of lipophilic medicines, small molecules, and proteins for the treatment of cancer, as well as potential advantages to carrier imaging moieties aiming at cancer theranostics [9,10].

Polymeric micelles (Figure 1) are prospective carriers for the distribution of many insoluble and poorly soluble pharmaceuticals that can be integrated into the hydrophobic core of the micelles due to these benefits and their tiny size (~100 nm) [11]. Moreover, polymeric micelles are thought to have advantages due to their strong core–shell structure and kinetic stability. The variety and adaptability of polymers that can be used to create micellar systems increase their potential for use in medication delivery applications. There have also been reports of polymers that can give polymeric micelles a stimuli-responsive nature in addition to those that form the core and corona [12]. The micellar structure’s stimulus sensitivity is influenced by a number of “environmental” factors (external or internal), including pH, redox, enzyme activity, hypoxia, light, and temperature [10]. To increase target-specific drug delivery and regulate the rate of drug release in the tumor microenvironment, micelles can be improved by manipulating their chemical structure, physicochemical properties, and stability under pertinent conditions. The micellar systems, on the other hand, respond to stimuli by rupturing their structure and thus releasing the medicines [10]. Drugs are released at the tumor’s precise site, which reduces off-target drug binding and adverse outcomes. The particular way that is used for medicine delivery depends on the types, drug loading and entrapment effectiveness. In spite of their confirmed versatility, polymeric micellar systems remain elusive to the market and only certain products are under clinical investigation or have reached clinical application.

The current review aims to give a general understanding of the polymeric micelles, new advances and several ways for delivering cargoes to tumor sites. Additionally, this review article will go into detail about their “intelligent” response to stimuli and delivery of drugs for the treatment of various cancers. Additionally, we also attempted to provide insights on the regulatory features and prospects for clinical translation.

## 2. General Issues and Status Quo of Polymeric Micelles

### 2.1. Polymeric Micelles and Micellar Structures

Polymeric micelles represent drug delivery nanosystems that possess a core–shell structure and are created when amphiphilic block copolymers (ABCs) self-assemble in an aqueous solution [13]. Polymeric micelles also refer to systems where the lipophilic portion of the amphiphilic polymer is directed to the center of the micelles and the hydrophilic portion is directed outward. The hydrophilic component of the amphiphilic polymer is directed toward the core in reverse micelles, while the lipophilic component is directed outwards [13]. In the case of reverse micelles, the lipophilic portion of the amphiphilic polymer is directed outward and the hydrophilic portion is directed toward the center. Solubilizers are added to the surfactant micelle to create mixed micelles. The lipid bilayer breaks when surfactants are added, resulting in the creation of mixed micelles that contain both surfactants and polar lipids [13]. 

The hydrophobic component of block copolymers is meant to expel the medicine from the polymeric micelles and solubilize poorly soluble pharmaceuticals in the core. Furthermore, hydrophobic interactions between the drug and the hydrophobic unit of copolymers are widely acknowledged as a key element in both, slowing the drug’s rate of release into the external solution and solubilizing it in the micelles [13]. Many hydrophobic copolymers have been created and tested as core-forming building blocks (Table 1), and they exhibit the ability to solubilize poorly soluble medicines. It is usual practice to encase the hydrophobic core of PMs with hydrophilic copolymers, such as (i) poly(oxazolines), (ii) poly(ethylene glycol) (PEG), (iii) chitosan, (iv) hyaluronic acid, and (v) dextran [13] (Table 1).

According to numerous research on the hydrophilic shells’ functions, the systemic circulation time, biodistribution, and stability of the micelles in vivo, are all directly correlated with the physicochemical characteristics of hydrophilic polymers, such as surface density and molecular weight.

There are multiple options for the formation of polymeric micelles, depending on the characteristics of the polymer and solution. As a result, diverse polymeric micelles can be produced via di-block, tri-block, and multi-block copolymers, graft polymers, stimuli-sensitive polymers, etc. [17]. Furthermore, the kind that is generated can be considerably influenced by the solvent, pH, polymer concentration and ratios, co-solvent, etc.

Block copolymers can be categorized into one of three groups based on the intermolecular forces that regulate the segregation of micelles in an aqueous environment. They are micelles produced by metal complexations, amphiphilic micelles (hydrophobic interactions), and polyion complex micelles (electrostatic interactions) [17,18]. Polymeric micelles are primarily divided into two categories based on the manner of drug encapsulation: either chemical covalent binding of pharmaceuticals or the physical encapsulation method.

Multiple medications can be trapped inside the micellar structure to increase the therapeutic efficacy of the nanosystems. Drug loading may be assisted by micelle polymer chemical conjugation or physical trapping. Chemically conjugated medications are typically released by surface erosive processes or the total breakdown of the micelles, whereas drugs loaded using a physical entrapment approach are typically released by simple diffusion [18].

The position of the drug molecules within the polymeric micelles is based on both the properties of the drug and the length of the polymer chain in the amphiphiles. Pharmaceuticals of intermediate polarity are enclosed between the core and shell as well as nonpolar medications on the core and polar drugs on the shell [14,19]. Therefore, the drug solubility, stability, and pharmacokinetic profile as well as stabilizing compounds that are degradable are improving. Drugs’ interactions with polymers might result from electrostatic interactions or from covalent bonding. This leads to various drug loading or encapsulation capacities in the micelles [14,19]. The hydrophobic component of block copolymers is intended to liberate the medicine from the micelles and dissolve poorly soluble pharmaceuticals in the core. Additionally, it is well-acknowledged that the hydrophobic interactions pharmaceuticals—hydrophobic units in copolymers have a significant role in both slowing the pace at which medications are released into external solutions and causing the drugs to become soluble in the polymeric micelles [15].

Amphiphilic invertible polymers (AIPs) represent a family of polymers that produce/self-assemble micellar systems considering the polymer structure and concentration. The ability of AIP macromolecules to undergo reversed conformational changes in response to shifting solvent polarity distinguishes them from other types of molecules. The invertibility of novel polymers is encouraging for applications requiring controlled self-assembly in polar and non-polar fluids, in particular, medication delivery [20]. AIPs are composed of macromolecules that are alternatively distributed in a macromolecular backbone from a precisely controlled number of hydrophilic and hydrophobic short fragments with a well-defined length. Compared to the structure of block copolymers, the incompatibility of these small macromolecular fragments causes microphase separation at smaller length scales. The latter, in turn, allows for more control during micellar formation [21]. The hydrophilic–lipophilic balance (HLB), which significantly impacts the surface activity and capacity for self-assembly in polar and non-polar solvents, distinguishes the amphiphilic invertible polymers from one another. The AIP micellar assemblies bind lipophilic and hydrophilic guest molecules in water and toluene, respectively, acting as a host for ordinarily insoluble compounds in polar (including aqueous) and non-polar solutions [21].

### 2.2. Critical Micelle Concentration

Generally made from ABCs at or above their critical micelle concentration (CMC), polymeric micelles are a class of colloidal dispersions. CMC primarily depicts the equilibrium between the hydrophobic and hydrophilic segments and defines the micelles’ thermodynamic stability. The CMC is influenced by the hydrophobic group properties, the hydrophilic component’s molecular weight, and the hydrophilic component’s distribution within the amphiphilic polymer [22]. 

The copolymers can self-assemble into the spherical core–shell shape polymeric micelles because these hydrophobic and hydrophilic segments locally phase separate in aqueous solution [14]. The micelles that are generated are thermodynamically stable as long as the concentration of amphiphilic polymers in the solution is higher than the CMC [14]. 

Micelles break down at a rate that is largely determined by the amphiphile structure and the interactions between the chains upon dilution to a concentration below the CMC [14]. Micellar structures’ low CMC (0.1–1 µM) is what gives them their clinical advantages [23]. To lower the system’s free energy, the hydrophobic copolymer blocks self-associate within the micelle center, away from the aqueous surroundings, during micellization. However, the hydrophobic blocks form a shell by being positioned between the core and the surrounding environment (or corona) [24]. These hydrophilic copolymers include poly(oxazolines), poly(ethylene glycol) (PEG), chitosan, hyaluronic acid (HA), and dextran [12,14]. The following hydrophobic copolymers are utilized in micellar systems: poly(caprolactone) (PCL), poly(lactide) (PLA), polyesters, lipids, and poly(lactide-co-glycolide) (PLGA) [12].

The characteristics of the hydrophilic and hydrophobic segments dictate which polymer should be used. The hydrophobic core should have a high loading capacity and be very compatible with the medicine that is included. Additionally, the CMC and, consequently, the stability of the micelle are determined by the kind and molecular weight of the hydrophobic block [22].

Lower CMC is caused by the core-forming polymer’s higher molecular weight and hydrophobicity. On the other hand, the hydrophilic corona needs to shield the micelle sterically and be biocompatible and biodegradable [23,24]. As it is water soluble, biocompatible, uncharged, and offers steric protection, polyethylene glycol (PEG) is a widely used polymer for the hydrophilic corona for these objectives. Additional choices are poly (N-isopropyl acrylamide, or pNIPAM), and poly (N-vinyl pyrrolidone) (PVP). Hydrophobic polyesters are the most often used materials for hydrophobic cores, however other materials, such as polyethers and polypeptides, are also employed. Examples of regularly used polymers are poly (propylene oxide) (PPO), poly (d,l-lactic acid) (PDLLA), poly (l-aspartate), and poloxamers [12,14,23,24].

### 2.3. Preparation Methods

Both chemical and physical routes can be used to create micelles. For the chemical approaches, a reversible chemical link hydrophobic functional ingredient—the amphiphilic polymer is created and the hydrophobic ingredient is then enclosed in the micelle’s core. For physically obtaining route, an amphiphilic polymer self-assembles into a core–shell structure micelle, in a solution, encasing a hydrophobic functional component in the core through hydrophobic interactions and/or hydrogen bonding. The benefits of the physical approaches are their simplicity and applicability to hydrophobic components. The synopsis of these techniques, together with their benefits, drawbacks, and applications, can be found in Table 2, while Figure 1 is dedicated to a schematic representation of these methods.

#### 2.3.1. Direct Dissolution

The self-assembly of amphiphilic polymers in aqueous solution is the most facile technique for creating polymer micelles. For polymers with good water solubility, the direct dissolution approach is the most appropriate. The polymer self-assembles into micelles with the help of gentle and continuous stirring in water when the polymer concentration is higher than the CMC [28]. Although the direct dissolution approach is an easy way to make micelles, poorly soluble compounds find it challenging to construct a stable micellar structure.

#### 2.3.2. Solvent Evaporation/Film Hydration

When the copolymers are soluble in a volatile and water-miscible organic solvent, thin-film hydration/solvent evaporation techniques are used. In terms of the timing of polymeric micelles production and solvent evaporation, thin-film hydration and evaporation slightly differ from one other. The thin-film hydration process involves dissolving copolymers in an organic solvent, evaporating the solvent to create a thin polymer film, adding a water phase to hydrate the film, and stirring to produce the micelle. Additionally, the solvent evaporation approach implies the dissolution of the copolymer in an organic solvent, adding water to the mixture to create the micelle, and then evaporating the solvent [29].

#### 2.3.3. Oil in Water Emulsion

This method represents a practical technique for creating micelles when the drug and copolymers are soluble in water-immiscible organic solvents (e.g., chloroform, dichloromethane, ethyl acetate, and methylene chloride). In the first step, hydrophobic functional components and the polymer are dissolved in organic solvents that are not water soluble; this represents the oil phase [30]. Then, to create an oil-in-water emulsion, they are gradually added to the aqueous solution while being vigorously stirred, homogenized, or both. The polymer rearranges to form a micelle structure, with the internal phase being an organic part and the external phase being a continuous aqueous phase. After that, the evaporation of the emulsion’s organic phase creates a micellar solution. 

#### 2.3.4. Dialysis 

One of the most popular ways to incorporate biologically active chemicals is via dialysis. The dialysis process is typically used to create micelles for functional compounds and/or polymers that are poorly water soluble. In general, the polymer and the material to be incorporated are moved from a solvent that is selective for the polymer’s hydrophilic chains to a solvent that is considered for the material to be embedded, such as deionized water [31]. The hydrophobic functional component is transported to the micelle core as the hydrophobic chains of the polymer gradually coalesce under the influence of a selected solvent, in order to form the micelle core. After extending the dialysis period for a few days, the organic solvent is totally eliminated. Organic solvents such as ethanol, acetone, dimethyl sulfoxide, and tetrahydrofuran are frequently employed.

The osmosis effect of the dialysis membrane, which is much stronger than the driving force of micelle creation in the direct dissolution process, is used in the dialysis method to prepare micelles via solvent exchange. Micelle size increases as a result of the incorporation of additional polymer molecular chains into micelles.

#### 2.3.5. Other Preparation Methods

Ultrasonic treatment is another technique for obtaining micelles. The encapsulation of poorly hydrophilic polymers and hydrophobic components is generally not accomplished with ultrasound alone, and the ultrasound-assisted approach is typically utilized for polymers with strong water solubility.

#### 2.3.6. Functionalization Methods

Proteins, peptides, nucleic acids, and phospholipids can all be chemically attached to or physically contained in the micelles to produce bio-functional polymeric micellar systems.

Generally speaking, biomolecules can either be integrated into the hydrophobic section (core) of the micelles or conjugated to their hydrophilic part (shell or corona), depending on their nature. 

The circulating time can be increased by isolating and re-dissolving micelles, resulting in stable NPs. The basic cross-linking strategy is the most susceptible. This can be accomplished by adding a polymerizable group to the block copolymer’s hydrophobic moiety or by introducing a polymerizable monomer to the micelle core that is then polymerized using a specific initiator [32,33,34]. The micelle core’s reduced free volume can occasionally have a negative impact on the capacity for drug loading. Similarly, the shell of micelles can be cross-linked and the core of the cross-linked shell micelle can disintegrate, producing nanocontainers.

Additionally, the micelles hydrophilic tail’s terminal end can be functionalized, offering the micelles a higher chance to function as nanocarriers. Cross-linked micelles maintain their structure even at concentrations below their CMC, which results in the formation of a polymeric amphiphile. The core cross-linking frequently improves micelle stability. The overall physicochemical and biological characteristics of the micelles are altered when the shell is functionalized by biomolecules, which results in the creation of novel nanocarriers for targeted drug delivery applications [32,33,34].

Functionalizing the chain ends of the soluble shells is another technique to change the micelle’s morphology or properties. The covalent connection between a chain end and a potential ligand is a part of chemical functionalization. The extremely specific ligand receptor binding aids in directing the release of the solubilized medicines to the desired area. For better solubilization and release, the inclusion of various chemicals and salts that can fine-tune micellization and micelle properties can be used. Furthermore, a core–shell aggregate made of a stimuli-responsive block copolymer can be employed to load drugs. Under the impact of outside stimuli such as pH, temperature, and magnetic response, the medication may be released [18].

### 2.4. Polymeric Micelle Types

Depending on their amphiphilic nature and the solvent parameters, such as type of solvent, polymer concentration, ionic strength, pH, etc., many types of polymeric micelles exist.

The construction of an amphiphilic block copolymer with the desired properties is the first stage in the creation of polymeric micelles. To generate spherical micelles as illustrated in Figure 2(I), the used block copolymers must have hydrophilic and hydrophobic block domains, with the length of the hydrophobic monomer being somewhat shorter than the hydrophilic monomer [18]. The co-polymers amphiphilic nature makes it easier for them to self-assemble into micellar forms [35]. 

Other differences in the hydrophilic and hydrophobic segment lengths may cause or even prevent the formation of aggregates with distinct morphologies [18]. Different types of polymeric micelles are created using A-B copolymers, A-B-A tri-block copolymers, and graft copolymers, where A stands for the hydrophilic block and B for the hydrophobic one (Figure 2(II)). As it is the most stable conformation when hydrophilic–hydrophobic interactions are at play, micelles are typically formed into spherical shapes.

Tetrablock and pentablock copolymer manufacturing has been reported, and starblock copolymers are also being researched [36,37,38]. In the latter, blocks are joined to a branching point with more than two polymers rather than being placed in a linear manner (Figure 2(II)) [38].

The micelles can also be classified by taking into consideration their morphology (disk-like, toroidal, and bicontinuous) and shapes (e.g., star-shaped, worm-like, flower-shaped) [39,40]. Additionally, polymeric micelles can be categorized as “smart” due to their reaction to their surroundings; they are pH- [41], temperature- [42], or light-responsive [43]. These types of micelles will be further discussed in the next sections.

Several reviews have gone into great detail about the polymeric micelle’s types and their corresponding features and properties [44,45,46]. Due to this aspect, we present the micelle kinds based on their functions (Figure 3) and make a brief description based on the context of our review.

#### 2.4.1. Conventional Polymeric Micelles

Taking into account the intermolecular forces that separate the core segments from the aqueous environment, there are three basic types of polymeric micelles: those produced by hydrophobic interactions, those produced by electrostatic interactions, and those produced by noncovalent interactions [47].

##### Polymeric Micelles Generated by Hydrophobic Contact

Hydrophobic interactions, which take place between the core and the shell structure in the aqueous medium, are the building blocks of polymeric micelles generated by hydrophobic contact [48]. Using techniques such as reversible addition-fragmentation chain transfer (RAFT) polymerization, the polymer hydrophobic lengths can be advantageously modified [49].

##### Polymeric Micelles Generated by Electrostatic Interactions

Electrostatic interactions between two oppositely charged moieties, such as polyelectrolytes, are the basis for the construction of polymeric micelles produced through polyionic interaction; they are also known as polyion complex micelles. In the presence of oppositely charged polymers, the corona of the micelles can be penetrated, resulting in the formation of the polyion complex micelles. The electrostatic and van der Waals forces that are present in polyion complex micelles can be changed in order to regulate the structure and size of the material [50]. The straightforward synthesis process, facile self-assembly in aqueous medium, structural stability, high drug loading capacity, and sustained blood circulation are just a few of the special characteristics of polyion complex micelles [51]. Poly(ethylene oxide)-b-poly(methacrylic acid) (PEO-b-PMA) [52] and PEG-chitosan [53] are two of the most popular amphiphilic block copolymers for creating polyion complex micelles.

##### Polymeric Micelles Generated by Noncovalent Interaction

Specific intermolecular interactions, such as hydrogen bonding, allow the core and shell of these polymeric micelles to be joined non-covalently at the ends of their homopolymer chains. These micelles are also known as “block-copolymer-free” because the preparation methods do not make use of block copolymers. Graft copolymers, homopolymers, and oligomers are the polymers that are employed to make these kinds of polymeric micelles. The most utilized amphiphilic polymers are PEG [54] and poly(4-vinylpyridine) [55].

#### 2.4.2. Functionalized Polymeric Micelles

Despite having many advantages, the sole polymeric micelles have certain drawbacks, chief among which is their inability to perform targeting at specific locations. Therefore, functionalized polymeric micelles have been created to address this issue and, in recent years, they have gained considerable attention for their ability to deliver medications to ill regions [56].

##### Cell-Penetrating Polymeric Micelles

The target of several medications, including DNA, siRNA, polypeptides, and oligonucleotides, is found inside the cells. Thus, a carrier needs high cellular transmembrane transportation. A potential method for achieving this goal is to functionalize a polymeric micelle surface with a cell-penetrating substance, such as a peptide. The way in which micelles penetrate different types of membranes and cells can be found in a recent review by Cai et al. [57]. Drug delivery with this type of micelle has included parenteral, nasal, and oral routes [25].

##### Targeting Polymeric Micelles

Polymeric micelles are successful nanocarriers for the administration of drugs because of their wide range of nanosizes and narrow size distribution, which reduces the danger of blood vessel occlusion and delays rapid drug removal [58]. They are able to deliver a variety of medications, including proteins, peptides, chemotherapy agents, antidiabetic agents, antituberculosis agents, siRNAs, and plasmid DNAs (pDNAs), among others, at their target areas.

The polymeric micelles can be altered by conjugating them with different ligands, such as antibody fragments, glycoproteins, transferrin and folate, in order to target the site precisely, without harming other healthy cells [59,60,61]. 

##### Active Targeting Polymeric Micelles

A drug-carrying carrier system is delivered to a specific region through surface modification as opposed to spontaneous reticuloendothelial system (RES) absorption [62]. Techniques for surface modification include applying a bioadhesive, non-ionic surfactant, monoclonal antibodies that target a particular cell or tissue, or albumin protein [63].

On several levels, active targeting can be adjusted in the following manner: 1. First order targeting (organ compartmentalization): only the capillary bed of a chosen target site, organ, or tissue receives drug carrier system distribution. 2. Second order targeting (cellular targeting): the precise administration of medicine to a particular cell type, such as cancer cells (without affecting the normal cells). 3. Drug distribution to the intracellular organelles of the target cells is known as third order targeting (intercellular organelles targeting) [64].

Active targeting can be accomplished in two ways: (i) by taking advantage of the disease’s biology; or (ii) by using outside stimuli or triggers. Designing actively targeted micelles has the goal of altering the ligand to boost their selectivity for tumor cells, increase intracellular delivery and accumulation, and lessen adverse effects and related toxicities [65].

Active targeting makes use of receptor-mediated endocytosis, in which the micelle-attached ligands interrelate with particular receptors (that are either overexpressed or exclusively expressed) present in the diseased tissue’s cell membrane, causing endocytosis and internalization. In the case of normal tissues, these receptors are either not expressed or their expression is very low [65]. The carrier can also be modulated such that it reacts to the pathological triggers particular to the condition. Typically, ligands are conjugated to the outer ends of the hydrophilic segment to modify polymeric micelles for active targeting [65]. Targeting ligands used to create polymeric micellar systems are typically categorized as (a) small molecular weight molecules (e.g., folic acid [66], sialic acid [67], biotin [68]); (b) antibodies and their fragments [69]; (c) peptides and proteins such as trans-activator of transcription (TAT) peptide [70] and arginine–glycine–aspartic acid (RGD) peptide [71]; and (d) aptamers [72]. Due to their high binding affinity and specificity for tumor cells, antibodies and peptides are proven to be superior ligands. However, antibodies are not broadly used because they are expensive and their production is highly complex, whereas peptides are thought to be less costly than antibodies. Aptamers have gained popularity as a targeting ligand in recent years due to their high stability, minimal immunogenicity, and ease of manufacture; nonetheless, their application is constrained due to nuclease degradation [73].

For improved therapeutic effectiveness, the modification by specific ligands makes the micellar systems valuable for reaching desired targets such as cancers. As an example, one method to target the glucose transporter 1 (GLUT1), which is overexpressed on vascular endothelial cells in most cancers, is to modify cisplatin-loaded polymeric micelles with glucose. The transcytosis of PMs crossing the blood-tumor barrier might be greatly facilitated by conjugating 25% of glucose to the PEG chains [74]. In comparison to the free medication and non-targeted micelles, the micelles modified with 25% glucose boosted tumor exposure in OSC-19 and U87MG xenografts by 2 and 10 times, respectively [74]. Arginine–glycine–aspartic–phenylalanine acid (RGDF) peptide modification significantly increased tumor cellular uptake efficiency via RGDF-mediated endocytosis in addition to reducing mononuclear phagocyte systems clearance and raising plasma AUC, and 6 wt.%-RGDF polymeric micelles significantly inhibited tumor growth in mice with the H22 tumor by 96% [75]. 

##### Passive Targeting Polymeric Micelles

Passive targeting is primarily accomplished by the increased permeability and retention effect (EPR), hypervascularization, and inadequate lymphatic drainage [10] that are brought on by the overexpression of vascular endothelial growth factor receptors [76].

Normal blood arteries have pores < 6 nm in size; however, malignant blood vessels have pores that are ~100–600 nm in size, which makes it easier for micelles with a particle size of ~10–30 nm to accumulate inside diseased cells. Further research revealed that larger particles that are ingested are easily absorbed by the RES; as a result, the particle size of the polymeric micelle must be <150 nm in order to effectively accumulate within cancer cells via the EPR effect and avoid RES detection. The polymeric micelle’s inclusion of amphiphilic block copolymers also creates a neutral environment that hinders RES’s ability to recognize it easily, enhancing passive targeting [77].

#### 2.4.3. Mucoadhesive and Mucus-Penetrating Polymeric Micelles

Mucus represents a complex viscoelastic fluid made up of glycoproteins, proteins, and polysaccharides, and is frequently found in the gastrointestinal tract, genitalia, lungs, and eyes [78].

According to the polymer properties, mucus-acting polymeric micelles are defined as having mechanical or physical interactions between the polymer chains and the mucus layers that cause the micelles to either become trapped in these layers (mucoadhesive) or penetrate the underlying tissues (mucus-penetrating) (Figure 4).

The most popular methods for creating these muco-acting polymers are functionalized polymers with muco-acting moieties and employing polysaccharides as a template for polymers [78].

#### 2.4.4. Stimuli-Responsive Polymeric Micelles

Due to their capacity to regulate drug release at specific sites while minimizing drug exposure in off-target sites, stimuli-responsive polymeric micelles have drawn a lot of attention [79]. The stimuli-responsive polymeric micelles can be positively and predictably managed by using external and/or internal stimulation or even pathological alterations in the target tissues as triggers. Enzyme-, thermo-, pH-, redox-, light-, and multi-responsive polymeric micelles are the six subgroups that commonly make up stimuli-responsive PMs. 

We will present the above-mentioned subclasses in the following section, in relation to their use in drug delivery applications.

### 2.5. Biological Barriers and Polymeric Micelles for Efficient Anticancer Therapeutic Drug Delivery

The effectiveness of cancer nanomedicine is generally assessed by the number of medicines that can reach the tumor site. Diverse biological barriers constitute both a difficulty and an opportunity for developing specialized medication delivery systems that can successfully reach the target region. The following section briefly presents various biological barriers, their peculiarities (Figure 5), and methods for going beyond them.

The efficacy of nanotherapeutics in disorders ranging from cancer to inflammation is constrained by obstacles to drug delivery. Shear pressure, protein adsorption, and rapid clearance are a few of the physiological and biological obstacles that need to be overcome by micellar systems for efficient biodistribution and drug administration [80]. These obstacles are frequently altered by disease conditions, making it harder to remove them using a tried and true method [81,82]. Such alterations of the biological barriers are challenging to discover and thoroughly characterize since they occur at the systemic, microenvironmental, and cellular levels, and vary from patient to patient.

The biodistribution and clearance governed by interdependent systems are one of the most difficult systemic hurdles facing the successful delivery of micellar systems. Structure and chemical mechanisms that guard against exposure to hazardous chemicals prevent the delivery of foreign compounds to the body. Due to first pass pulmonary absorption in the case of lung malignancies, inhalation or intravenous treatment are preferred with particles larger than 100 nm [83]. Moreover, the circulatory system provides both size restriction and ongoing immunological monitoring, depending on anatomical location, of the basal and endothelial membranes. 

Identifying the biological barriers at organ and cellular levels that patients confront as well as on a case-by-case basis enables the development of the best polymeric micellar platforms. Site-specific drug delivery will remain an elusive target until nanocarrier design has addressed the majority of the biological barriers met upon administration. Although nanomedicine and nanodelivery systems are emerging fields, overcoming these barriers and incorporating unique design elements will lead to the development of a new generation of nanotherapeutics, marking a change in basic assumptions about polymeric micellar-based drug delivery.

#### 2.5.1. Systemic Barriers

The selective localization of nanostructures can be influenced by the endothelium and basal membranes’ varying pore sizes, which are dependent on the anatomical location. For instance, the blood arteries within the bone cavity have significant gaps between endothelial cells and a discontinuous basal membrane, both of which encourage the deposition of nanoparticles. However, unlike the adrenals, the lungs and endocrine glands have a continuous basal membrane with somewhat fenestrated endothelial cells, which leads to a smaller concentration of particles of a similar size. The localization of nanostructures based on size is determined by the cumulative impact of endothelial pore size [84].

The specific characteristics of the tumor vasculature have an impact on the distribution and delivery of micelles. Angiogenesis, a dynamic process that aids the tumor development, increases the availability of oxygen and other nutrients, enabling cell proliferation and tumor growth. This process requires the release of signaling molecules, which include proteins such as vascular endothelial growth factor (VEGF-A) [85]. Over-secretion of VEGF promotes rapid angiogenesis, because of its unregulated speed in the growth of leaky vasculature with higher permeability [86]. The enhanced permeability and retention (EPR) effect, which is caused by the “leaky” vasculature, allows for nanostructures accumulation in the tumor [87].

Nanostructures delivery to tumors is nevertheless minimal despite this unusual pattern of distribution, demonstrating that the EPR effect is insufficient to ensure micelles accumulation and activity on their own [87]. The idea that the design can enable endothelial transcytosis, offering a different channel to the tumor, is one unique strategy to circumvent the dependence of delivery on EPR [88]. Research on the sizes of nanostructures [88] and surface alterations with ligands for vascular and/or tumor-expressing receptors has shown encouraging results in terms of enhanced internalization and transcytosis [89].

The body’s clearance processes pose a further obstacle to the micelle’s delivery and retention. Rapid clearance lessens the nanostructures buildup and activity at the target site, even though clearance is a crucial component of delivery for clinical usage [90]. The main organs for micelles elimination are the liver, spleen, and kidney [57]. Modulating size and surface features can prevent quick clearance by these organs and lengthen the circulation half-life inside the body, as will be covered in the following section. To extend the circulation period, some polymers are coated with polyethylene glycol (PEG), a technique that has proven highly effective [57].

#### 2.5.2. Organ-Level Barriers

There are other hurdles based on the tumor niche in addition to the mononuclear phagocytic system (MPS), which makes up a significant portion of the RES and hinders the spread of micellar systems. The distribution and uptake of micellar systems face numerous problems due to organ-specific architecture and resultant vascular permeability, even though PEGylation has been proven to prolong the circulation time and allow escape from being cleared by the MPS and RES [57].

The blood–brain barrier (BBB), which tightly controls how much of the brain is exposed to the systemic environment, serves as one example. The steady rate of bad outcomes in brain cancer patients demonstrates the difficulty of removing this barrier. The brain side of the BBB is lined by brain capillary endothelial cells (BCECs), which are highly polarized and have functionally separate luminal and abluminal membrane compartments [91]. These cells differ from endothelial cells present in peripheral tissues, which are responsible for the majority of the BBB’s selective capabilities. Tight junctions (TJs) at the lateral, luminal membrane connect BCECs instead of extensive fenestrations, which offer a high-resistance barrier to the passage of tiny hydrophilic molecules and ions [92].

The most prevalent primary brain tumors (intra-axial) in adults with significant heterogeneity are gliomas [93]. They primarily have neuroepithelial origins and have various mutational profiles in different patients [93]. According to the type of glial cells implicated in the tumor, these are made up of astrocytomas, oligodendrogliomas, and ependymomas and share characteristics with the glial cells of the brain [93]. Glioblastoma stem cells, which are extremely invasive, aggressive, and therapy-resistant, make up the glioma cells. They are thus distinguished by an invasive phenotype with strong migratory potential [94]. As a result, it is claimed that encapsulating various treatments into polymeric micelles will increase drug transport across the BBB. For example, Meng et al. obtained a 0.9-folds higher drug penetration across BBB by functionalized micelles than non-functionalized ones [95]. In another study, a functionalized polymeric micellar system co-loaded with Anti-BCL-2 siRNA and temozolomide reduced the tumor volume in rats and the expression level of BCL-2 in glioma cells in comparison to functionalized micelles containing individual therapy [96]. By integrating disulfide links (reduction-responsive) into the polyurethane backbone coupled with pH-sensitivity (PMeOx), Zhang et al. created polyoxazoline-polyurethane (PMeOxPU(SS)-PMeOx) based polymeric micelles for the effective administration of doxorubicin to glioma cells. The in vitro investigation found that dual responsive functionalized polymeric micelles released drugs 1.2 times more readily at pH 5.0 than they did at pH 7.4 buffer in the presence of the redox reagent dithiothreitol. Due to the fact that C6-glioma cells were not found to be toxic to the generated dual-responsive blank polymeric micelles, they are an appropriate nanocarrier for in vivo drug administration [54].

#### 2.5.3. Cellular-Level Barriers

Moving the micellar systems into the tumor cells once they have arrived at the target organ, is quite difficult. The micelles are cell internalized by means of phagocytosis, macropinocytosis, receptor-, caveolin-, clathrin-, or endocytosis-mediated endocytosis, as well as transcytosis. 

The majority of tumor cells use either clathrin- or caveolin-mediated cellular endocytosis as their primary endocytosis mechanism. Other cell types have different operational endocytosis processes, and changes in the extracellular environment have an impact on these routes [97]. Since most nanostructures tend to cluster or agglomerate in biological fluids, resulting in changes in size, it is crucial to understand how the targeted cell interacts with its environment while developing nanostructures [97].

Molecules ≤ 60 nm can undergo caveolin-mediated endocytosis, which uses lipid rafts to form specialized vesicles following entrapment [98]. Nanorods-shaped micellar nanostructures are more likely to undergo this type of endocytosis than nanospheres, which are often taken up by clathrin [99]. Clathrin-mediated endocytosis relies on receptor-mediated, hydrophobic or electrostatic contacts between nanostructures and the cell membrane in regions of clathrin expression [98,100]. It is the most frequent method for nanostructures uptake in non-specialized mammalian cells. 

The endocytic pathways’ activation is regulated by the nanostructure’s characteristics such as rigidity and size. Although there are some variations in the results, thicker ones are typically more readily ingested, and both experimental and theoretical assessments suggest that the endocytosis of rigid particles takes less energy [97,101]. Furthermore, too-small (≤30 nm) nanostructures might not be able to drive membrane wrapping sufficiently to initiate endocytic processes [86]. When particles with a diameter of <50 nm are utilized, excellent cellular absorption and intracellular delivery are reported in numerous studies [86,102].

There are only a few clear trends regarding the ideal shape and size of nanostructures for the subsequent delivery phase of cell uptake [86,103]. Nonetheless, certain models and studies suggest that spherical-shaped nanostructures have enhanced uptake over rod-shaped ones in non-phagocytic cells [86,103], but other studies prove otherwise [104]. As a result, a variety of parameters, such as the properties of the cell membrane and those of the micellar systems, which also have an impact on the subsequent endocytic process, dictate the method employed for micelles absorption. 

## 3. Polymeric Micellar Systems with “Smart” Responsiveness

Stimuli-sensitive systems have been in the focus of the scientific community for quite some time due to the fact that they can release their cargo at a selected action site and at a specific time [105]. To obtain the best therapeutic targeting, there should be no drug escape from the polymeric micelle during circulation. After the micelles have built up in the targeted tissue, the medicine should not be released. It should only be released on the application of an internal or external trigger. It is to note that the prolonged release of drug payloads from polymeric micelles in the circulation or stable confinement of drug load within micellar cores are prerequisites for the realization of targeted drug delivery. In the first scenario, pharmacological payloads have congruent pharmacokinetics profiles and follow a trajectory comparable to that of polymeric micelles. In the latter scenario, medications are progressively released from polymeric micelles; unlike encapsulated medications, only “free medications” are able to produce both pharmacological and toxicological effects [57].

Stimuli-sensitive polymeric micelles are produced by increasing the potential of these adaptable drug carriers by exploiting diseased or stimulus-induced alterations in the target tissues as triggers. Most diseased areas, especially cancerous ones, vary from healthy tissues in terms of pH, redox potential, temperature, expressed enzyme profiles, and oxygen saturation levels. These variables are known as “internal stimuli” due to the fact that they act as local triggers. The use of heat, ultrasound (US), near-infrared light (NIR), or magnetic fields are examples of “external stimuli” [106] (Figure 6).

The stimuli-triggered reaction in the polymers can cause the micelles to disintegrate, destabilize, isomerize, polymerize, or agglomerate [107]. The micellar system relies on healthy-sick tissue differences to release drugs. The term “intelligent delivery,” could also be used to describe this kind of release mechanism. Premature drug release can be avoided thanks to their responsiveness [108], which signaled a trend in research towards stimuli-sensitive drug delivery systems. A combination of multiple or more stimuli-sensitive systems has also been seen to enhance targeting effectiveness.

The next section of this review will concentrate on different stimuli-sensitive micellar delivery methods for treating cancer, including reduction-sensitive, pH, thermo-responsive, and ultrasound-sensitive ones.

### 3.1. Internal Stimuli-Responsive Polymeric Micelles

Internal/biological stimuli, which are largely specific to ill tissues (e.g., cancerous tissue) can be exploited to increase the drug’s action specificity, and to build stimuli-sensitive drug delivery systems (Figure 7). The choice of an appropriate material is essential when creating nanocarriers because of the presence of oxidative stress [109], an acidic or neutral environment in tissues [110], varying concentrations of glutathione (GSH) in the cytoplasm and nuclei [109], overexpressed enzymes such as matrix metalloproteases (MMPs), metabolic enzymes in lysosomes, and increased production of reactive oxygen species (ROS) in the mitochondria. These particularities can lead to the rupture of the structures of polymeric micelles, resulting in the release of the encapsulated medication at the desired sites.

#### 3.1.1. pH

There are numerous characteristics that are utilized as significant “signatures” because the metabolic profile of tumors differs from that of normal tissues. For instance, the high amounts of lactic acid brought on by inadequate oxygen perfusion cause the pH value in the tumor microenvironment to be typically lower than that in normal areas [111]. Normal tissues and blood have an extracellular pH of 7.4, but malignancies have a pH range of 6.0–6.5 [46]. Researchers may be able to make use of this in healthy–malignant tissues pH variation in endosomal and lysosomal compartments as an internal stimulator for triggered chemotherapeutic medication release [46]. Due to the existence of an ionic block or an acid-labile link, pH-sensitive PMs can be distorted to aid the release of medications in moderately acidic circumstances outside or inside the tumor cells because they are stable at physiological pH conditions [46]. The EPR effect of micelles could be used to regulate the hydrophobic agent release in tumor tissue.

Most multifunctional drug delivery systems use pH-sensitive micelles that react to the low pH stimuli of the tumor microenvironment [112]. The use of pH-based degradable linkers is one method for determining the pH sensitivity (acetals or hydrazones). These chemical compounds link the hydrophobic and hydrophilic polymeric micelle-building subunits together. Micelles easily degrade the connections between the hydrophobic and hydrophilic molecules when they arrive at the low pH tumor microenvironment [112]. 

In another method, one can make use of polymeric building blocks that are sensitive to the pH, such as poly(-amino ester) and poly(amino acids), which will undergo charge conversion in response to a low pH stimulus, altering the structure of the polymer and leading to drug release [112].

Table 3 summarizes various pH-responsive polymeric micellar systems employed for the treatment and management of different types of cancers; here we included the polymers used, the drugs entrapped, and their therapeutic outcomes.

It is interesting to note that an acidic tumor microenvironment can also be employed to increase the internalization of co-loaded drugs. In a novel drug delivery system (THCD-NPs), HA-conjugated curcumin (Cur) and d-α-tocopherol acid polyethylene glycolsuccinate (TPGS) were used as selective drug-carrying vehicles to deliver dasatinib (DAS) to liver cancer cells for combined administration [121]. The micellar system was pH sensitive and could accelerate drug release at low pH conditions. In vitro experiments exhibited a significant cytotoxic effect on HepG2 cells. In vivo experiments showed that THCD-NPs significantly inhibited tumor growth and reduced the toxic side effects of free drugs in a mouse solid tumor model [121]. Cur and 5-fluorouracil (5-FU) were both loaded into the poly(2-vinyl pyridine)-b-poly(ethylene oxide) (P2VP_90_-b-PEO_398_) block copolymer in order to create “intelligent” delivery systems [122]. The in vitro drug release at pH 2, 6.8, and 7.4 revealed that the pH regulates the release of both medicines. The drug release efficiencies surpass 90% at pH 2. Additionally, in vitro tests showed that these micelles are hemocompatible and biocompatible [122]. An amphiphilic block copolymer of monomethoxyl poly(ethylene glycol), poly(l-lysine), and poly(aspartyl(Benzylamine-co-(Diisopropylamino)ethylamine)) mPEG-PLLys-PLAsp(BzA-co-DIP), abbreviated as PELABD, was synthesized to prepare a pH-responsive micelle to co-deliver hydrophobic DOX and siRNA into cancer cells [123]. In vitro experiments indicated that the size of micelles could be changed at different pH values, and DOX could be released rapidly at low pH values. The viabilities of Bel 7402 cells decreased with the increase in DOX concentration [123]. 

#### 3.1.2. Temperature

One of the stimuli with the most comprehensive investigation for cancer medication delivery applications is temperature. Some polymers are completely soluble and in an extended state above a certain temperature, known as the lower critical solution temperature (LCST) [107]. The term “LCST” refers to the phase transition temperature (PTT) below which polymers are water-soluble and above which they are water-insoluble [44]. The main mechanism of the micellar systems is that as they move through heated tumor tissue, where local temperatures rise over their LCST; the micelles’ outer shells change into hydrophobic structures that can bind to and enter cells by means of hydrophobic contact. Due to this, anticancer drug-loaded micelles amass at malignant tissues and kill cancerous cells. 

The thermoresponsive polymer poly(N-isopropyl acrylamide) (pNIPAAm) (LCST at 32 °C) is the most often utilized kind to obtain polymeric micellar systems that respond to the temperature of diseased tissues. pNIPAAm is reported to experience a reversible phase shift in an aqueous medium [107]. For example, to be used in the structure of the drug carrier platform, a thermoresponsive PNIPAAm-b-PLA amphiphilic block copolymer micelle was synthesized by Ghasemi et al. [124]. The temperature-sensitive polymeric micelle-loaded DTX underwent a temperature-prompted phase shift as a result of heating near its LCST in the as-prepared state. Due to the temperature difference between cancer cells and normal cells, this property was leveraged to preferentially absorb PNIPAAm-b-PLA/DTX into tumor cells. The DTX delivery at 40 °C was made by means of the carrier’s thermo-responsiveness property. As a result, PNIPAAm-b-PLA loaded DTX exhibits significant promise as a highly effective anti-cancer agent [124]. 

Poly(N-vinylcaprolactam (PNVCL), poly(N-vinylisobutyramide, poly(N-vinyl-n-butyramide), poly(2-isopropyl-2-oxazoline, and poly [2-(2-ethoxy)ethoxyethoxyethyl vinyl ether] [105,125] are other thermosensitive synthetic polymers.

#### 3.1.3. Redox

Due to a considerable difference in glutathione (GSH) content between the tumor and the normal tissue micro-environment, reduction potential had been a useful biomarker for designing polymeric micellar systems, which rely on redox-responsive triggers. In general, the GSH concentration in the tumors is of ~four times higher than in the microenvironment of normal tissues [126]. By considering GSH, it may be possible to boost medication release from redox-sensitive polymer carriers inside tumor cells while reducing release from the cell to the surrounding tissue [110].

Disulfide bonds (-SS-) are covalent bonds created when two thiol groups are conjugated; these connections are easily broken by a specific GSH concentration and are typically included in different systems to achieve redox sensitivity; also, after bond cleavage, the medication is liberated in the cytoplasm and nucleus (Figure 8).

Many micelles are made from polymers with disulfide connections between the hydrophobic and hydrophilic segments that are redox sensitive [127]. These micelles go through reductive breakdown or decomposition to release medicines in a redox environment [128].

Meng et al. covalently bound medicines and polymers using disulfide-containing linkers. In this study, linkers were utilized to join Cur and a multivalent mPEG-polylysine copolymer. The linkers used were 3,30-dithiodipropionic acid (DDPA) and 4,40-dithiodibutyric acid (DDBA) (mPEG-PLys). As a result, the two amphiphilic conjugates (mPEG-PLys (Cur)-R4 and mPEG-PLys (Cur)-R5) that were produced could self-assemble into micelles. The disulfide bonds (-SS-) in DDPA and DDBA suffered reductive degradation, which led to the micelles’ disintegration and drug release, as shown by in vitro release tests. The mPEG-PLys (Cur)-R5 micelle had the highest level of cytotoxicity on HeLa, PC3, and 4T1 studied cell lines. Similar findings were demonstrated in nude mice bearing tumors, where the mPEG-PLys (Cur)-R5 group displayed a superior tumor suppressive effect [129].

Methoxy poly(ethyleneglycol)-b-poly(ε-caprolactone-co-α-azido-ε-caprolactone)(mPEG-b-poly(εCL-co-αN3εCL)) with reduction-sensitive bis(alkyne) crosslinking loaded with methotrex (MTX) was designed by Gulfam et al. In contrast to noncrosslinked micelles and free MTX, experimental studies showed that crosslinked polymers loaded with MTX have regulated drug release and better apoptotic capability against breast cancer cell lines (MCF-7) [130].

For the delivery of DOX, self-assembling, redox-sensitive polymeric micelles were created. The polymeric micelles also contained ibuprofen that was disulfide-bonded to the hyaluronic acid backbone and is redox-sensitive (HA-ss-BF). Excess GSH causes the disulfide link to dissolve and micelles to disassemble in the tumor microenvironment, releasing the medication and BF (anti-inflammatory drug). Breast cancer mouse models’ in vivo research indicated a good biodistribution and cellular absorption by tumor cells [131].

An amphiphilic block polymer self-assembled reduction sensitive micelle, mPEG-SS-PzLL/TPGS/DOX, was synthesized for efficient anticancer therapy. The polymer had a biodegradable backbone and disulfide bond can be cleaved by reduced GHS in tumor cells, which led to fast release of the DOX. TPGS was designed for an increasing drug accumulation and a reduction in drug efflux. In follow-up research, we found that mPEG-SS-PzLL/TPGS/DOX micelles achieved a high encapsulation efficiency of 96.1. The mPEG-SS-PzLL/TPGS/DOX micelles caused stronger cytotoxicity to 4 T1 cells and promising therapeutic efficacy for BALB/c mice bearing 4 T1 tumors [132].

In another study, novel redox-sensitive micellar systems were fabricated from the poly(caprolactone) conjugates with disulfide-linked PEG (DDMAT- mPEG-S-S-PCL, DPSP). Docetaxel (DTX) was employed as a model drug and encapsulated into the DPSP. The in vitro anti-tumor activity of the DTX-loaded DPSP and free DTX against the breast cancer cells (4T1) was evaluated by MTT assay. These systems displayed redox-responsive performance in the presence of GHS. Animal experiments indicated that the DPSP showed excellent blood compatibility and good bio-security. Cell tests suggested that the DPSP could be taken in by 4T1 cells smoothly, which improved the anti-tumor activity of free DTX [133].

Ibrahim et.al. produced amphiphilic block copolymer prodrugs, which are denoted as PEG-b-P(CPTM-co- ImOAMA) and consist of PEG and a copolymerized block of redox-responsive disulfide bonds-connected camptothecin (CPT) prodrug monomer (CPTM) and 1-(1H-imidazole-4-yl)-2-(octylamino)-2-oxoethyl methacrylate (ImOAMA) [134]. PEG-b- P(CPTM-co-ImOAMA), a block copolymer prodrug, may self-assemble in an aqueous solution to create core–shell polymeric micelles. For endosomal escape, the PImOAMA segments were included. The GSH-responsive cleavage of the disulfide linker from CPTM after translocation into the cytoplasm can hasten the release of active CPT by a self-immolating mechanism. The effective endosomal escape capabilities of the micelles and faster CPT medication release were confirmed by in vitro studies. Therapeutic outcomes were markedly improved by antitumor efficacy. This design outlines a practical method for enhancing the therapeutic efficacy of reduction-responsive disulfide bond-linked block copolymer prodrugs, which may enable block copolymer prodrugs to be more widely used in clinical settings [134].

#### 3.1.4. Enzyme

Enzymes are essential for many biological processes occurring within the body due to the fact that they are exceptionally specialized and have outstanding catalytic characteristics. Several diseases have been shown to have dysregulated enzyme activity. Proteases, peptidases, and lipases, which are crucial for tumor cell proliferation, angiogenesis, invasion, and metastasis, are frequently expressed at elevated levels in solid tumors compared to healthy tissues [135,136]

It is interesting to note that dysregulated enzymes have also been proposed as viable biological catalysts for individualized cancer therapy. Enzymes may catalyze reactions in benign conditions, and have several advantages when used as triggers. Enzymes can also catalyze certain chemical processes and have substrate selectivity. Various enzyme-triggered drug delivery systems, such as enzyme-responsive polymeric micelles, have been created for precise and efficacious targeted drug delivery [135,136]. Additionally, prodrug-based nanocarriers make use of enzymes to deliver drugs intracellularly and boost drug action. The potential of polymeric micellar systems with specific moieties in their backbone or side groups that were selectively identified and destroyed by overexpressed enzymes in the extracellular/intracellular milieu of malignancies is investigated. As a result, hydrophobic medicines can now be delivered to tumors while minimizing the negative effects in healthy organs [137].

Extracellular matrix (ECM) remodeling proteases are known as MMPs and are included as members of the calcium- and zinc-dependent family of endopeptidases. MMPs play a critical role in tumor invasion and metastasis by destroying the histological barrier of tumor cell and nearly all protein components in ECM [138]. MMPs, in particular MMP2 and MMP9, are overexpressed in tumoral microenvironment, as compared to healthy tissues. Thus, many drug delivery systems had been fruitfully adjusted using MMPs to increase tumor penetration and/or tumor targeting [139]. In one investigation, a PEG (M_ñ_5000)-PLA block copolymer with the MMP-responsive peptide GPLGVRGDG was investigated [66]. The peptide GPLGVRGDG was sensitive to MMP2 up-regulation at the tumor location, and MMP2 specifically degraded the peptide between the amino acids G and V. As the morphology of the micelles did not change considerably, the release rate of PTX in the presence of MMP2 was not significantly different from that in the absence of MMP2 [140]. In a different work, PTX was encapsulated using a copolymer made of PEG2000-MMP2-sensitive peptide (pp)-trans-activating transcriptional activator (TAT) (GPLGIAGQYGRKKRRQRRRC)-PEG1000-phosphoethanolamine (PE). The micelle’s outer layer was created by the PEG2000, which also served as a chain of defense. The middle layer that targets tumors was created by the pp, a linker. The inner core of the micelles and the middle layer that penetrates cells were created by TAT peptide and PEG-PE, respectively. Under the catalysis of MMP2 in the ECM, the bond of the MMP2-sensitive peptide was broken, and the PEG shell was eliminated. The MMP2-sensitive micelles increased tumor penetration and micellar cargo uptake, according to an in vivo tumor retention trial [141]. In a recent study, a polymeric, enzyme-responsive, and biodegradable micelle for the targeted delivery of cabazitaxel cancer was designed [142]. Two amphiphilic block copolymers were used for the micelle’s generation: PEG, an enzyme-responsive peptide, and cholesterol make up the first block copolymer, while a targeting ligand, PEG, and cholesterol make up the second. The enzyme-responsive peptide is cleavable in the presence of MMP2, which is overexpressed in the tumor microenvironment of prostate cancer. The micelle significantly outperformed free cabazitaxel in terms of cellular absorption in prostate cancer cells. In contrast to unmodified micelle and free cabazitaxel, the ligand-coupled polymeric micelle showed superior tumor growth suppression in mice harboring prostate cancer xenografts [142].

Additionally, it is possible to take advantage of the activity of specific enzymes, including NAD(P)H:quinone oxidoreductase-1 (NQO1), that are present in particular cancer microenvironments [143]. NQO1 is an overexpressed enzyme in certain tumor types; it maintains homeostasis and impedes oxidative stress caused by elevated reactive oxygen species (ROS) in tumor cells [143]. An amphiphilic block copolymer (QPA-P), composed of NQO1 enzyme-triggered depolymerizable QPA-locked polycaprolactone (PCL) and poly(ethylene glycol) (PEG) as hydrophobic and hydrophilic constituents, was synthesized. In aqueous settings, this QPA-P produced self-assembled micelles. It was found that NQO1 facilitated the depolymerization of QPA-locked PCL by a cascade two-step cyclization process, which ultimately caused the separation of micellar structure and prompted the release of loaded medicines at the target cancer cells. The NQO1-responsive micelle demonstrated higher anticancer effects and NQO1-triggered intracellular drug release when compared to the control group. These findings suggest that an anticancer drug delivery system using NQO1-responsive polymeric micelles has a promising future for enhancing treatment efficacy [144].

#### 3.1.5. Hypoxia

Cancer, cardiomyopathy, ischemia, rheumatoid arthritis, and vascular illnesses have all been linked to hypoxia (low oxygen levels) [145]. Lowered oxygen partial pressure enables tumor-specific medication delivery because hypoxic and normoxic cells are characterize by dissimilar microenvironments [145]. The tumoral microenvironments have significantly less oxygen than normal tissues, as seen by the oxygen-sensitive sensors [145]. There is proof that persistent hypoxia changes the biology of tumors and long-term hypoxia changes their pathophysiology. This includes choosing genotypes that promote survival under hypoxic rearrangement damage and altering genes expression to promote survival by reducing apoptosis, enhancing tumor angiogenesis, and encouraging autophagy [145].

In a recent study, nitrobenzyl chloroformate (NBCF) and chitosan (CS) were conjugated to create the chitosan-nitrobenzyl chloroformate conjugate (CS-NBCF) [146]. While no burst release occurs in normoxic settings, the drug DOX is rapidly released from the DOX/CS-NBCF in hypoxic conditions, reaching 68% within 24 h. CS-NBCF was discovered to be less damaging to NH-3T3 cells, demonstrating its potential to serve as an anticancer drug carrier and providing fresh ideas for the investigation of controlled drug delivery systems. In another report, a bone-targeted and hypoxia-responsive polymeric micelle system was prepared for the successful treatment of bone metastatic prostate cancer. The micelles had a high affinity for metastatic bone and a high sensitivity to responding to hypoxia in vitro. They were self-assembled from a polyethylene glycol and poly-l-lysine-based copolymer employing alendronate as a bone-targeted moiety and azobenzene as a hypoxia-responsive linker. Additionally, in vivo investigations have shown that the micelles may react to hypoxic bone metastases for rapid drug release to an efficient therapeutic dosage following a selective buildup in metastatic bone. By suppressing osteoclast activity and boosting osteoblast activity, the micelles might prevent bone degradation and tumor growth in bone, improving the therapeutic result [147]. By using hypoxia-induced drug release, an amphiphilic polypeptide was employed to self-assemble and deliver cytochrome C to specific tumor cells. The amphiphilic polymer employed was the hypoxia-responsive methoxy PEG-block-poly (diethylenetriamine-4-nitrobenzyl chloroformate)-l-glutamate. The system was transformed into a hypoxia-responsive polymer by the conjugation of 4-nitrobenzyl chloroformate. HepG2 liver cancer cells were able to demonstrate the micelles’ efficacious absorption. The cytochrome c-loaded micelles demonstrated good cytotoxicity in HepG2 cells under hypoxic circumstances [148]. The work by Lu et al. was motivated by the enantiomeric nature of poly(d,l-lactide) and created stereocomplex PEG-PLA micelles through stereoselective interactions of enantiomeric PLA [149]. To maximize therapeutic effects, these micelles were further combined with a hypoxia-responsive moiety used as a hypoxia-cleavable linker of PEG and PLA. The outcomes demonstrated that the produced micelles exhibited great structural stability, enhanced drug loading, and efficient drug delivery to tissues including malignancies. In particular, they have the ability to sensitively react to the hypoxic tumor environment for drug release, reverse hypoxia-induced drug resistance, and accelerate cell migration for increased bioavailability under hypoxia. The micelles, especially at a high dose, prevented the main tumor from growing and enhanced the pathological conditions of the tumor, which significantly reduced the spread of the tumor to the lungs and liver without generating any systemic damage, according to additional in vivo studies. Thus, hypoxia-responsive stereocomplex micelles become a trustworthy method of drug administration for the treatment of breast cancer metastases [149].

#### 3.1.6. ROS

ROS-responsive polymer nanoparticles regulate the release of a specific drug by taking advantage of the substantial ROS accumulation in some disease tissues. ROS, an oxygen-derived chemical species produced by the body, at low concentrations modifies cell signaling pathways and promotes cell proliferation [150]. The balance between ROS and antioxidants and their implications on tumor progression has been represented pictorially in Figure 9.

The endoplasmic reticulum and mitochondria both create ROS. ROS can generate non-specific protein damage and nucleotides such as DNA that can worsen the disease’s underlying causes if they are produced in excess [151]. For tumor-specific treatment, numerous functional moieties could be incorporated into the micelles to elicit ROS sensitivity in the tumoral microenvironment. This could help the redox-sensitive micelles to disassemble in the redox tumor microenvironment, thereby releasing the loaded entities. In order to create ROS responsive polymeric micelles, sulfur-containing linkers such as thioether, thioketal, and vinyl thioether have been used [151,152]. ROS oxidants can cleave these sulfur-containing linkers. Incorporating alternative linkers that are reactive oxygen species-responsive, such as polyproline, boronic ester, and selenium/tellurium, is an option in addition to sulfur-containing linkages. “Oxidative stress” results from a rise in ROS concentration that makes antioxidants (such as catalase or superoxide dismutase) inefficient at reducing it [153].

Since cancer cells overexpress ROS, researchers have developed ROS-sensitive micellar systems for targeted therapy. Accordingly, the goal of a recent research paper is to co-deliver the PLK1 inhibitor volasertib (BI6727) and the miR-34a mimic volasertib utilizing PEG-poly[aspartamidoethyl(p-boronobenzyl)diethylammonium bromide] (PEG-B-PAEBEA). With a 10% drug loading of volasertib, this polymer was able to self-assemble into micelles of about 100 nm and form a complex with miR-34a at a N/P ratio of 18 and higher. Volasertib and miR-34a treatment together demonstrated synergistic antiproliferative action, improved G2/M phase arrest, and inhibition of colony formation, which ultimately led to cell death. After systemic injection of micelles containing volasertib and miR-34a, respectively, orthotopic pancreatic tumor-bearing mice were examined. The tumor volume had significantly decreased, and a histological evaluation of the major organs revealed that there had been very little systemic damage. In conclusion, volasertib and miR-34a mimic-containing PEG-B-PAEBEA micelles may be used to treat pancreatic cancer [154].

In another study, palmitoyl ascorbate (PA), as a prooxidant for hydrogen peroxide (H_2_O_2_) production in tumor tissue was used to obtain a H_2_O_2_-responsive camptothecin polymer prodrug micelle, which offered the nanocarriers with self-sufficing H_2_O_2_ stimuli in tumor tissues, to achieve novel synergistic oxidation-chemotherapy [155]. By increasing tumoral H_2_O_2_ levels, tumor-specific H_2_O_2_ synthesis by the PA component overcame the intrinsic tumoral low ROS concentration and enabled high-efficiency in vivo tumor-targeted ROS-responsive drug release. The functional components were combined into a single nanoscaled delivery vehicle, as opposed to the straightforward mixing of Vitamin C and molecular anticancer medications, to address the pharmacokinetics discrepancy between Vitamin C and anticancer drugs. Finally, innovative synergistic oxidation-chemotherapy was found to have powerful in vivo anticancer efficacy via the obtained micelles and through tumor-specific H_2_O_2_ generation, which resulted in enhanced oxidative stress and responsive fast CPT release in the tumors. The benefits of the multifunctional cooperative system portend the likelihood that they will be applied in clinical trials. Due to this fact, this study serves as a proof-of-concept for responsive nanocarriers with self-sufficing properties through the active adjustment of tumor microenvironments, which should be highlighted as a novel conceptual perspective in promoting the applications of stimuli-responsive nanocarriers as anticancer drug delivery vehicles [155].

#### 3.1.7. Stimuli-Responsive Inversion of Macromolecules

It is possible to use AIPs’ capacity to bind lipophilic insoluble molecules in a solution and to change conformation in response to environmental polarity to investigate the interactions between AIP micellar assemblies and biomembranes in the hopes of developing them into novel nanopharmaceuticals.

It would be expected that the AIP conformation would change upon the adsorption of drug-loaded micellar assemblies onto the cell membrane, which is essentially a lipid matrix made up of a phospholipid bilayer with inwardly directed hydrophobic hydrocarbons and outward-oriented hydrophilic heads. The contact between the polymer and membrane may be improved as a result, leading to the release of the drug molecules [20,21]. The amount of the transferred medication is mostly determined by the micellar loading capacity; however, micellar assemblies have successfully conveyed the majority of the loaded cargo molecules across the polar–non-polar interface. Additionally, the therapeutic substance can be delivered into the interior of the cell without impairing its biological activity thanks to the inversion of the AIP and breakup of the carrier-drug complexes triggered by the interaction with cell membranes [21,156].

In a study, Polunin et al. [157] described a novel AIP-based micellar formulation of SN-38 and assessed its efficacy on growth inhibition in neuroblastoma (NB) cells obtained either at diagnosis or at relapse following extensive chemoradiotherapy. An AIP composed of alternating blocks of PEG and polytetrahydrofuran (PTHF) was used to create colloidally stable, drug-loaded micellar assemblies with a uniform size of ~100 nm (PEG600-PTHF650). Even after a brief (10 min) exposure, the micellar medication completely reduced the proliferation of chemo-naive NB cells when administered at low nanomolar concentrations (10–50 nM). Additionally, prolonging the exposure to 24 h caused the micellar formulation to have a significant and long-lasting inhibitory effect on the proliferation of NB cells demonstrating an acquired lack of p53 function [157].

The creation and characterization of an amphiphilic polymer with a hydrophobic palmitoyl (Pal) group and a zwitterionic poly(2-methacryloyloxyethyl phosphorylcholine) (pMPC) block that can form micelles and serve as a drug delivery system for hydrophobic anticancer medications such as doxorubicin (DOX) was recently described [158]. The research group postulated that the pronounced polarity contrast between the Pal domain and the pMPC block would strengthen the micelles and increase their capacity to hold drugs, while the pMPC shells would increase the stability of the micelles and the effectiveness of cellular uptake. The micelle formation, cytotoxicity, and drug loading of DOX of the Pal-pMPC and the Pal-PEG polymer were studied and compared. The cellular uptake and anticancer effects of the DOX-loaded Pal-pMPC micelles were further examined in cell culture systems using the multidrug-resistant AT3B-1 cell line and the non-multidrug-resistant HeLa cell line. The findings demonstrated the Pal-pMPC polymer’s low toxicity. In comparison to micelles created by the Pal-PEG polymer, the Pal-pMPC micelles showed a greater drug loading capacity and improved cellular internalization efficiency. Additionally, it was demonstrated that in an environment containing fetal bovine serum, multidrug-resistant cancer cells responded more effectively to DOX-loaded Pal-pMPC micelles than to Pal-PEG micelles in terms of fighting cancer [158].

### 3.2. External Stimuli-Responsive Polymeric Micelles

External stimuli have also attracted a lot of interest in the delivery of tumor-targeted drugs in addition to endogenous stimuli. Exogenous stimulation can be controlled in terms of its timing, location, and intensity compared to endogenous stimulus. In the past few decades, a great deal of research has been conducted on various stimuli, including light, temperature, electric field, magnetic field, and ultrasound. Each stimulus has unique properties and benefits. Light has drawn a lot of attention because it is a relatively common environmental stimulus [159]. For medication delivery systems, light offers a number of benefits, including non-intrusiveness, high spatial resolution, time control, convenience, and ease of use [160]. Another alluring stimulus is temperature since thermo-responsive nanomaterials can target pre-selected locations when the temperature changes [161].

#### 3.2.1. Light/Photo/NIR

Light-responsive micellar systems have the benefit over pH- or redox-responsive ones that no additional (bio)chemical additives are needed to trigger the responses in the view of causing controlled release of encapsulated molecules [162]. The use of ultraviolet (UV) and near-infrared (NIR) light can cause the release of medication from the micellar systems. Due to skin absorption, UV radiation cannot breach far into the body; nonetheless, it can be utilized to initiate the medication release from micellar systems topical therapies. NIR (λ  =  650–900 nm) is able to penetrate deeply into the body and is suitable as a systemic trigger [56]. However, due to their low phototoxicity, NIR-responsive micellar systems are favored over visible- and UV-responsive ones [12]. When exposed to light, these systems go through a photoisomerization reaction and an irreversible cleavage (photochemical phase transition). The hydrophobic block is changed into a hydrophilic block as a result, which causes disruption of the micelle and cargo release at the cancer site [163]. When exposed to NIR radiation, the micellar systems separate due to the presence of the NIR-responsive moiety and the loading of a photosensitizer. This happens as a result of the NIR-responsive moiety’s photo-oxidation reaction with the micelles ROS, which causes a fast medication release at the tumor location [164].

Typically, light-sensitive micellar systems are synthesized by incorporating and conjugating chromophores into the polymeric structure, such as azobenzene, pyrene, cinnamoyl, spirobenzopyran, or nitrobenzyl groups [165]. The irradiation with external light can regulate how quickly the medications that are encapsulated are released. Upon illumination, the light-responsive micelle’s structure changes, and the payload is released.

Siboro et al. report a systematic investigation of the effect of NIR light exposure on the PEG-*block*-poly(styrene-alt-maleic anhydride) polymeric micelles loaded with indocyanine green (ICG). The results displayed that the ROS produced from ICG not only reacted with diselenide bonds, but also attacked PEG chains, resulting in the polymer degradation (continuous even for 36 h after NIR exposure) [166].

Wei et al. created new light-responsive polymeric micellar system based on acetylated chondroitin sulphate (AC-CS) and protoporphyrin IX (PpIX) for encapsulating DOX and apatinib (Apa) concurrently. Due to the speedy cracking of the intelligent carrier and the rapid release of DOX and Apa caused by the photoconversion of PpIX to ROS under the influence of 635-nm red light, in vitro studies have shown that chemotherapy and photodynamic therapy work together to successfully treat tumor MDR. As a result, the cleavage of the polymeric micelle by localized illumination could significantly improve the safety profile of DOX/Apa by preventing the release and buildup of multiple drugs in areas other than tumors [167].

In another study, a light-responsive DOX conjugated polymer (poly-DOX) was developed for highly operative chemotherapy. A break in the DOX-PEG amide bond upon UV irradiation can cause PEG shed and increased DOX cellular uptake. This study aimed at demonstrating the effectiveness of Poly-DOX–micelle for anti-cancer, cellular uptake, and biosafety [168].

#### 3.2.2. Temperature

In the medication delivery systems, the temperature could be used either as an internal or external stimuli. The tumor tissue has a slightly raised temperature than the surrounding normal tissue, and this difference in temperature can be boosted by external heating (using an external heat source/device). The release of drug carriers, which are intended for sustained release over a particular area of tissue, is accelerated by an increase in temperature brought on by an external stimulation. Patients receiving chemotherapy frequently go through this process in addition to being made hyperthermic for thermal-based therapy [169]. An external stimulus-based strategy is more promising than an internal stimulus-based one, which is harder to manage due to a narrow temperature range.

Lower critical solution temperature (LCST) and higher critical solution temperature (UCST) polymers are two categories of thermo-responsive polymers [170]. Examples on thermos-sensitive polymers include poly(N-isopropylacrylamide) (pNIPAAm), poly(hydroxypropyl methacrylamide-lactate) (p(HPMAm-Lacn), and Pluronic^®^, [171]. The physical characteristics of thermo-responsive polymeric micelles, which contain thermos-sensitive blocks, alter dramatically as the temperature changes.

In a recent work, micelles loaded with camptothecin and DOX, which are prepared by a thermosensitive amphiphilic block copolymers (ABC), (P(HEMA-co-DMA)-b-P(AAm-co-AN) (PPy), which due to thermo-sensitive polymer swelling, caused by the hydrophobic to hydrophilic conversion, PPy released heat when exposed to NIR, activating the simultaneous therapeutic activation of PTT and chemotherapy [172].

#### 3.2.3. Ultrasound

In order to increase pluronic micelles for intracellular uptake, high-frequency ultrasound (US) causes medications to release from micelles in around 15–20s [173], which helps to prevent the cellular damage associated with radiation. Through the cavitational phenomena, they produce thermal and mechanical effects that lead to nanocarrier destabilization and drug release [107].

Ultrasonic waves are produced in tissues at ultrasound frequencies > 20 kHz. Tissues are impacted by US in two different ways: (i) by inducing heat, or (ii) by oscillating bubbles having a non-thermal effect. The medicine can penetrate the membrane more deeply and without harm thanks to the US. The medication accumulates through passive diffusion once the carrier is broken, which causes pores to form in the cell membrane [174]. The power density, concentration, and the nature (hydrophilic or hydrophobic) of the medicine and the process used for the US, are necessary to be taken into account for the careful application of US. Studies have shown that low frequency US usually penetrates deeper bodily tissues; as a result, it is used to treat cancers that are either deeply embedded or located in the distant region of the body [175].

Microbubbles or other ultrasonic contrast compounds are frequently used to promote US-induced cavitation and concomitant medication release. Ultrasonic-sensitive polymeric micellar systems are first gathered in the tumor site for cavitation of microbubbles, and then focused US is applied to speed up drug release and boost cellular absorption. [107].

Examples of micelles that have been extensively researched for US-triggered drug release include those based on PEO and PPO ternary copolymers [165].

Wu and his coworkers created pluronic P123/F127 micellar systems containing Cur and employing focused-US. Compared to free Cur, Cur-loaded micelles circulated for a longer time and took up more cells. Targeted ultrasonography after systemic injection of Cur-micelles indicated time-dependent tumor-targeting deposition. Significantly less tumor growth and weight resulted from the sonoporation-assisted site-specific chemotherapy, which raised the possibility that an US-guided nanomedicine deposit and local release could increase chemotherapy effectiveness [176]. In another study, Liu et al. prepared ROS-responsive micelles based on PEG-poly(propylene sulfide) (PEG-PPS) for targeted delivery and in situ drug release. Upon the irradiation with US, the loaded sonosensitizer hypocrellin (HC) generated ROS to trigger the disassembly of the micelles and meanwhile realizing sonodynamic therapy against cancer. The in vivo experiment indicates that the HC loaded PEG-PPS are biocompatible and much more efficacious than an equivalent amount of free HC in inhibiting the growth of cancer [177].

#### 3.2.4. Magnetic

There is a growing interest for using magnetic field-responsive anticancer medicines in micelles [178]. To produce magnetic sensitivity, magnesium oxide (MgO), magnetite (Fe_3_O_4_), and maghemite (Fe_3_O_3_) NPs are frequently used [165]. They are frequently referred to as superparamagnetic iron oxide NPs (SPIONs) due to their high superparamagnetism and small size. They can be attracted by/to a magnetic field, but their attraction disappears when the field is removed [165]. A magnetic field that can penetrate the human body’s tissue is used in magnetic resonance imaging (MRI) [165]. The research suggests that magnetic field-induced hyperthermia and magnetic field-guided drug targeting are the two processes underlying controlled drug release [179]. Magnetic micellar systems for medication delivery that use hyperthermia have been thoroughly studied. Local heat can both delay tumor growth and create imaging possibilities because of the magnetic response [165].

Using a Pd(II) complex catalyst as an initiator, phenyl isocyanide was polymerized to create amphiphilic thermos-responsive block helical poly(phenyl isocyanide). The polymer was capable of forming spherical micelles by itself. The magnetic complex micelles were created after Fe_3_O_4_ NPs were loaded. The magnetic complex micelles showed clear magnetic hyperthermia and reversible thermo-responsiveness. Studies on cell viability showed that the nanomaterials were both nontoxic and well-biocompatible. By adjusting the temperature, the magnetic complex micelles might be employed as nanocarriers to achieve the controlled release of DOX. As a result of magnetic hyperthermia caused by micelle-loaded Fe_3_O_4_ NPs and efficient drug release caused by the morphology change of thermos-responsive poly(phenyl isocyanide)s, intracellular experiments showed that the magnetic complex micelles demonstrated excellent anticancer synergistic thermo-chemotherapy performance under an alternating magnetic field [180]. A ferrimagnetic PEG-poly(2-hexoxy-2-oxo-1,3,2-dioxaphospholane) (mPEG-*b*-PHEP) copolymer micelle loaded with hydrophobic iron oxide nanocubes and emodin was recently developed [181]. This composite micelle with a flowable core exhibited a quick response to magnetic hyperthermia, leading to an alternating magnetic field-activated supersensitive drug release. With the high magnetic response, thermal sensitivity and magnetic targeting, this supersensitive ferrimagnetic micellar system accomplished an above 70% tumor cell killing effect at an extremely low dosage, and the tumors on mice were entirely eradicated [181].

### 3.3. Dual/Multi-Responsive Micellar Systems

A variety of multifunctional stimuli-responsive polymeric micellar systems have been created and produced for medication targeted administration and controlled release in the treatment of cancer, drawing inspiration from all the particular properties of the malignancies microenvironment. By combining multiple sensitivities into one micellar system, a precisely controlled drug distribution and release can be achieved, resulting in greater in vitro and/or in vivo therapeutic activity. These systems include, but are not limited to, dual pH/redox-, dual pH/thermo-, dual redox/enzyme- and multiple pH/thermo/redox-responsive. In Table 4, we introduced some examples of dual/multiple responsive micellar systems.

## 4. The Use of Polymeric Micellar Systems to Develop (Bio)Sensors

### 4.1. General Aspects of (Bio)Sensors

In recent years, there has been a lot of focus on monitoring and controlling numerous and various parameters in fields such as clinical diagnostics, hygiene, environment, the food industry, forensics, research, etc. Therefore, it is necessary to have trustworthy analytical tools that can carry out precise analyses quickly. In addition, escalating worries over the general populace’s exposure to dangerous substances have spurred the imperative need for creating and building novel sensing and detection equipment. Biosensors are a type of technology that may be able to satisfy the demands mentioned above. In order to detect chemical molecules, a biosensor often uses isolated enzymes, immune systems, tissues, entire cells, and organelles as mediators of specific biochemical reactions [199].

Typically, biosensors are made up of an electronic system, a component known as the transducer, and a (bio)recognition element commonly referred to as the bioreceptor (often combined with the transducer) (Figure 10). The main components of biosensor technology are bioreceptors, which are biological molecular species that rely on biochemical mechanisms for recognition. The binding of interesting analytes results in a signal that the transducer can measure thanks to bioreceptors [200].

Biosensors can be classified as optical, thermal, piezoelectric, quartz crystal microbalance, or electrochemical depending on the type of transducer they utilize. Additionally, there are conductometric, amperometric, and potentiometric electrochemical biosensors [201].

Based on the biorecognition basis, there are two main categories of biosensors: (i) an affinity biosensor, which is typical of DNA and antibodies [202], and (ii) a catalytic biosensor, which is typical of enzyme biosensors [203]. As a result, a biosensor using an electrochemical transduction mechanism and an enzyme bioreceptor may be referred to as an enzyme biosensor or a catalytic biosensor (based on the biorecognition principle) [204]. They are also referred to as enzyme-based electrochemical biosensors since they are transducer- and bioreceptor-based. The specific enzymes utilized as bioreceptors in enzyme biosensors can also be used to classify them (glucose biosensor, urea biosensor, cholesterol biosensor, etc.) [201]. DNA biosensors (DNA as a bioreceptor) and immunosensors are examples of further biosensors (antibody as bioreceptor) [205].

The main analytical advantages of biosensors technology are adaptability, portability, high sensitivity, intrinsic selectivity, and simplicity to use in moderately complex environments due to their quick response.

Fluorescence sensing platforms [206] and electrochemical sensors [207] are just a couple of the new biochemical techniques and sensors that have emerged as a result of extensive research into detection technologies.

The sensitive fluorescence-quenching property of micellar systems forms the basis of fluorescence sensing. Because micelles have a large surface area and useful properties, they have a high affinity and selectivity for the analyte, which is the basis for electrochemical sensing [208]. Contrarily, colorimetric sensing relies on a color change that manifests as a result of the breakdown of micellar systems brought on by analyte recognition when the right ligand is used.

### 4.2. (Bio)Sensing Applications Based on Polymeric Micellar Systems

Due to their functional characteristics, particularly their high surface-to-volume ratio, size-tunable property, low toxicity, biocompatibility, prolonged circulation time, and specially designed recognition sites for the target analyte, polymeric micelles are appealing candidates for the fabrication of sensors. Additionally, they benefit from stability, sensitivity, and low production costs [209].

Micellar systems have proved effective for sensing applied in the detection of a variety of analytes, including metal ions, contaminants, and biomolecules such as protein, glucose, and urea as well as enzymes [210]. These sensing platforms are made by taking into account the detection goal and different micelle characteristics and can be also divided into colorimetric-, electrochemical-, and fluorescence-based sensors.

Drug and contaminant detection, and quantification are essential in a variety of fields, including forensics, the pharmaceutical industry (where a drug’s quality is evaluated all through the manufacturing process to ensure the correct effectiveness, tolerability and safety), the environment (such as in waste waters), and clinical toxicology. This subject is of crucial importance for economic and public health reasons, and it also appears on international political agendas [211]. Consequently, there is currently a lot of research being conducted on developing sensitive, quick, and affordable approaches. Due to their versatility, speed, portability, affordability, and ease of customization for drug detection, electrochemical sensing platforms are a favored solution for this type of application. However, the ability to miniaturize and portability of electrochemical sensors is the main benefit, allowing for the creation of smaller devices.

Electrochemiluminescence (ECL) sensor technology [212] for antibiotic detection has advanced quickly in comparison to fluorescence and electrochemical ones due to its benefits of strong controllability, good selectivity, high sensitivity, and simple instrumentation. It is widely known that a sensor platform’s capacity for detection is directly influenced by the characteristics of ECL reagents.

For the sensitive detection of ciprofloxacin (CFX), a novel molecularly imprinted sensor based on an aggregation-induced ECL reagent was created [213]. The electrode surface was changed to include nanozymatic ferriferrous oxide@platinum NPs (Fe_3_O_4_@Pt NPs) as a signal amplification element. A MIP was then created on the altered electrode utilizing CFX as a template molecule. The sensitivity of the sensor was greatly enhanced by the aggregation-induced luminescence effect and nanozyme amplification, whereas the MIP effectively improved the selectivity for CFX. In untreated milk samples, CFX recoveries of 92.0–111% were achieved. Therefore, the developed sensor exhibited good reproducibility, stability, and selectivity for CFX detection [213].

The use of electroactive molecularly imprinted polymeric micellar (MIP) nanoparticles enabled the development of a highly sensitive disposable electrochemical sensor for paracetamol (nanoMIPs) [214]. Solid-phase synthesis was used to create nanoMIPs. Itaconic acid served as a particular functional monomer in the polymer composition, while ferrocene served as a redox label to give the NPs electroactivity. The surface plasmon resonance studies performed by Alanazi et al. supported the nanoMIPs’ strong affinity for paracetamol [214]. The sensor performed exceptionally well and exhibited no cross-reactivity with interfering chemicals (caffeine, procainamide or ethyl 4-aminobenzoate). Its usefulness for point-of-care diagnostic applications was demonstrated by the sensors’ high reproducibility (RSD, 4.8%), quick reaction time (8 s), and reasonable shelf life (90 days) [214].

Another research group designed a novel electrochemical sensor by modifying a L-histidine functionalized multi-walled carbon nanotubes (L-His-MWCNT)-molecularly imprinted polymer for the discriminating, fast, and facile detection of tetracycline (TET) [215]. PDMS-1000 was decorated with pristine multi-walled carbon nanotubes (P-MWCNTs) by physical adsorption and then functionalized with L-histidine (L-His-MWCNTs@PDMS-5). The MIP-based electrochemical sensor was built on a glassy carbon electrode (GCE) by sol-gel consisting of tetraethoxysilane (TEOS) and cetyltrimethylammonium bromide (CTAB) with L-His-MWCNTs@PDMS-5 in the presence of TET [215].

Veterinary drug residues in milk can be found using fluorescence sensors; however, they are typically masked by the complex milk matrix. DNA stabilized silver nanoclusters (DNA-AgNCs), which emit light in the visible red wavelength region, can be created using a certain DNA template. By avoiding the fluorescence signal produced by the natural component of milk in the relatively short wavelength region, will lessen the milk fluorescence interference. The red fluorescence also has a reasonably high penetration, which can minimize the influence of interference-causing compounds. Therefore, Wang et al. developed a fluorescent sensor based on fluorescence resonance energy transfer (FRET) to identify kanamycin (KAN) residue in milk [216]. The designed sensor has acceptable specificity and sensitivity for detecting KAN residues. The applied strategy showed the limit of KAN detection to be as low as 22.6 nM [216].

Nonsteroidal anti-inflammatory drugs (NSAIDs) are widely used over-the-counter drugs and their uncontrolled disposal is a significant environmental concern [217]. Even natural antibodies have difficulty telling NSAIDs apart due to their great structural similarities [217]. Despite this, the molecularly imprinted cross-linked micelles created by Duan et al. showed outstanding affinity for and ability to recognize these medicines. It took less than two days to create and purify MIP-based fluorescence sensors for various medications. According to the binding investigations, these sensors could bind the desired medication only when compared to analogues, and they could also categorize the analogues according to how structurally similar they were to the original template. Fluorescent sensing could quickly find 50 ng/mL or 100–200 nM of the medicines in water.

Zhou et al. prepared fluorescent micelles as promising platforms for rapid detection of aliphatic amines in water [218]. A polyglycidol-b-poly(ethylene glycol)-b-polyglycidol triblock copolymer (GEG) was chosen because of its good water solubility, favorable interaction with amines, and simplicity of manufacture. Due to its high fluorescence quantum yield and solid-state fluorescence characteristics, a typical aggregation induced emission (AIE) dye of tetraphenylethene (TPE) was chosen as a reporting unit. Due to the activation of the limitation of intramolecular movements (RIM), TPE was released effectively in the solid state, but only faintly in the solution state. A luminous amphiphilic block copolymer was produced when the TPE moiety was selectively attached to PG segments (GEG-TPE). At an 8 g/L concentration, GEG-TPE self-assembled into micelles and made it possible to detect aliphatic amines in water in the order of seconds [218].

Based on a capture-report technique, Zhou et al. developed a new class of swellable fluorescent micelles for the quick and accurate detection of hazardous aromatic pollutants (APs) in water [219]. An aggregation-induced emission (AIE) chromophore-capped amphiphilic triblock copolymer self-assembled in an aqueous solution into core–shell micelles. Block polymer hydrophobic segments were arranged into cores along the core/shell interface with the AIE chromophore, which were supported by a corona of water-soluble polymer segments. The water-soluble polymer segments successfully collected APs. The hydrophobic cores of micelles were then filled with the captured contaminants. After absorbing APs, the cores grew larger, which caused the AIE chromophores’ fluorescence to be quenched. With a concentration of 1 ug/L, the fluorescent micelles enabled the quick detection of APs. The fluorescent micelles are substantially faster and use much less of the sample than commercial GC-MS while maintaining the same sensitivity [219].

Some critical components for both the environmental and biological systems are represented by different ions, which must be present under controlled conditions. An ongoing problem is the presence of excess ions as well as an ion deficit in biological (blood serum and plant parts) and environmental (groundwater) systems.

In order to specifically detect Al^3+^ and Fe^3+^ ions in environmental and biological samples, Han et al. created different micellar systems made of poly (ethylene oxide)-b-poly (N-isopropyl acrylamide-co-2,4-methacryloyl benzaldehyde oxime) and poly (ethylene oxide)-b-poly (N-isopropyl acrylamide-co-rhodamine 6G methyl acrylic acid) [220]. The fluorescence of other common metal ions, such as K^+^, Na^+^, Li^+^, Ca^2+^, Sr^2+^, Ba^2+^, Co^2+^, Fe^2+^, Ni^2+^, Mn^2+^, Mg^2+^, Cd^2+^, Cu^2+^, Zn^2+^, and Cr^3+^, did not significantly change in this micellar system. They also provided a simultaneous amount detection of Al^3+^ and Fe^3+^ ions, higher water solubility, increased detection sensitivity, and excellent biocompatibility. More interest has been paid to this than the single response probe because it might be employed as a multi-metal ion fluorescence chemosensor [220].

Halder et al. created a nano-sensor called FeNSOR employing a neutral micelle called TX-100 that has been properly porphyrin sensitized. Using a digital Pi-camera and related hardware and software, we have also created an instrument named “FeNSOR Device” (Iron sensor) based on the produced nano-sensor. The system operates according to the fluorescence spectrophotometry theory. The effectiveness of the nano-sensor and the device is found to be in the sub-molar range, with minimal interference from other ions, for the detection and estimation of iron ions in groundwater as well as in human blood serum. It was discovered that the threshold for detection and the standard deviation of the mean were 0.07 M and 0.016 M, respectively. The designed device is simple to use and has better repeatability and sensitivity than those that are currently on the market [221].

## 5. Clinical Status of Polymeric Micelles

Polymeric micelles are an excellent delivery strategy with higher translational efficacy since they have amazing qualities such as improved tumor targeting. Several polymeric medical devices are now being evaluated clinically, but the majority are yet to be evaluated pre-clinically. In Table 5, we collected different micellar products successfully marketed in the last years or under different stages of clinical trials.

Genexol^®^-PM is undergoing a phase II clinical trial in the United States and has received clinical application approval in South Korea, Hungary, and Bulgaria. Genexol^®^ represents a copolymeric micelle made from mPEG-PDLLA that is designed to boost PTX therapeutic effectiveness and water solubility. This micellar formulation can be used to treat non-small cell lung cancer, metastatic breast cancer, ovarian cancer, as well as other types of cancer [223,224] (p. 2). In comparison with Taxol^®^, which is formulated with 50 % ethanol and an equal amount of Cremophor EL, Genexol^®^- PM uses PEG-PDLLA to replace Cremophor EL, thereby accommodating more PTX and avoiding severe toxicities linked to systemic dosing of Cremophor EL [222]. In a phase II clinical trial for the treatment of metastatic breast cancer, Genexol^®^-PM led to a higher response rate when compared to Taxol^®^ (58.5 vs. 21–54%) [222]. Genexol^®^-PM in combination with gemcitabine demonstrated a higher overall response rate (46.5%) than Taxol^®^ (27.5 and 31%) in combination with gemcitabine in a different phase II clinical trial for advanced non-small cell lung cancer [222].

In comparison to Taxotere^®^, Nanoxel^®^ M, a docetaxel formulation based on mPEG-PDLLA, exhibits better safety profiles by lowering the incidence of taxane-induced peripheral neuropathy in breast cancer patients, fluid retention in animal studies, and Tween 80-associated hypersensitivity reactions [222]. However, in human lung tumor xenografts, Nanoxel^®^ M only exhibits antitumor effectiveness, which is comparable to Taxotere^®^ [222]. The treatment of MBC, NSCLC, ovarian cancer, and Kaposi’s sarcoma with Nanoxel^®^, a PTX-loaded micellar formulation made of the pH-sensitive block copolymer poly(vinylpyrrolidone)-block-poly(N-isopropyl acrylamide) (PVP-b-PNIPAM), has been approved in India [222]. Nanoxel^®^ demonstrated improved treatment efficacy in comparison to Taxol^®^ in a phase II randomized research for advanced breast cancer, with an overall response rate of 40 vs. 31% [222]. Unfortunately, the comparisons between Nanoxel^®^ and conventional formulations have been hampered by the design flaws and/or inadequate sizes of clinical studies, which have hampered its approval in other nations [222].

Additionally, there are more micellar formulations undergoing clinical trials than ever before. Some examples include those containing PTX (NK105), DOX (SP1049C, NK911), SN-38 (NK012), epirubicin (NC-6300), cisplatin (NC-6004), oxaliplatin (NC-4016), and docetaxel (CriPec^®^), but unfortunately, many of them only show only moderate efficacy in addition to improved tolerability [222].

NK 105, a micellar system made of 4-phenyl-1-butanol and PEG-poly (aspartic acid) copolymer, was created in order to boost irinotecan’s (topoisomerase-I inhibitor) water solubility. It underwent phase I of a clinical trial where the pharmacodynamics, toxicity, and suggested dose were reported. Furthermore, in phase II, 57 patients participated in the clinical investigation. Next, a phase III clinical trial was conducted; the drug’s therapeutic efficiency was compared to that of PCX-treated metastatic breast cancer patients [19]. NK105 showed inferiority in median progression-free survival when compared to Taxol^®^ (8.4 vs. 8.5 months) [222].

Another micellar system created to boost the efficiency of irinotecan hydrochloride is NK 012. This micelle also contains a poly (l-glutamic acid) fragment of PEG–P(Glu) block copolymer. Phase II clinical trials for NK 012 micelles have just begun and it has successfully completed two clinical trials performed in the USA and Japan [223].

Epirubicin was conjugated with PEG-poly (α, β-aspartic acid) via a hydrazone bond in NC 6300 to create a pH-sensitive polymeric micelle. Phase I/II clinical trials on NC 6300 were conducted to determine the recommended dose, safety, and tolerability in patients with recurrent solid tumors [222].

NC 6004 was formulated by using PEG-b-poly (l-glutamic acid) in order to change the cisplatin’s pharmacokinetic profile and bio-distribution within solid tumors. Phase I trials using this micellar system and gemcitabine are concurrently conducted on patients with pancreatic cancer. The combination had positive phase I/II results, which prompted a comparison with free gemcitabine in phase III clinical trial [222].

DACH-platinum is a component of NC 4016 polymeric micelles, which were developed to treat advanced solid tumors with improved blood plasma circulation. Phase I of the clinical trial for NC 4016 is complete [222].

## 6. Advantages and Drawbacks of Polymeric Micellar Systems

Micelles’ core–shell structure is their primary distinguishing feature. The corona shell protects the drug by limiting its removal by the mononuclear phagocyte system (MPS), allowing for longer blood circulation [225]. As a result, hydrophobic medicines can nevertheless be stable in water. Micelles can also be eliminated by renal filtration and are less harmful [226]. Drugs that are insoluble in water can be stored and derived from micelles in their hydrophobic core [226]. The best micelles for delivering hydrophobic medications were those with a hydrophilic corona to stabilize and shield the hydrophobic medication. Medicines can be made 10- to 500-fold more water soluble by being enclosed in polymeric micelle moieties [227], allowing for the intravenous injection of hydrophobic drugs that are micelle-encapsulated. Regarding the drawbacks of polymeric micellar systems, different research groups have created a number of strategies to circumvent the difficulties that arise when using them. A brief overview of the tactics that might be employed to reduce the potential limitations of micellar systems is given in Table 6.

## 7. Conclusions and Outlooks

The global burden of cancer is rising quickly every day and the therapeutic choices offered by traditional drug delivery systems appear to be constrained as a result of several problems. Poor oral bioavailability, low dissolution rate, and lack of site specificity are a few of them. Therefore, it is being investigated whether standard treatment procedures might be modified to overcome these obstacles. Advances in theranostics were made possible by nanotechnology and nanomedicine. Due to the functional characteristic of block copolymers, polymeric micellar systems have attracted interest as potential nanocarriers during the past few years. In comparison to different vesicles, micellar systems are nanosized, monodispersed, reasonably stable, and inexpensive. By modifying their chemical structure, the latter can also be surface-modified or stimuli-sensitized.

Understanding the crucial obtaining process variables that each technique involves will give researchers a better starting point for customizing the formation of polymeric micellar systems with desired physicochemical features. This could eventually result in their commercialization for clinical use and the creation of cutting-edge treatments. Polymeric micellar systems play the role of nanocarriers in a variety of biomedical applications due to their physicochemical characteristics, including their nano size, EPR effect, low CMC, and stimuli-responsiveness. Polymeric micellar systems are actively researched as theranostic nanocarriers, are commercially available in cosmetics, and numerous clinical trials have been carried out with encouraging outcomes due to the above-mentioned advantages.

Currently, there are still many obstacles to overcome before micellar formulations can be used in clinical settings. The biggest obstacle is the discrepancy between the therapeutic efficacy in preclinical models and clinical studies. Concerns that polymeric micelles might simply function as solubilizers rather than delivery systems are raised by the fact that most preparations either demonstrate minor improvement or similar efficacy to the standard of care. Systemic investigations are highly needed to show the in vivo fate of polymeric micelles, including, but not limited to, the process of entry into the body.

Polymeric micelles do, however, provide significant potential for being used in biomedical applications. It is possible for them to be successfully positioned in the market for a variety of biomedical applications by overcoming the difficulties of drug loading, looking for opportunities for their scale-up, and conducting a thorough examination of their fate in biological systems.

## Figures and Tables

**Figure 1 pharmaceutics-15-00976-f001:**
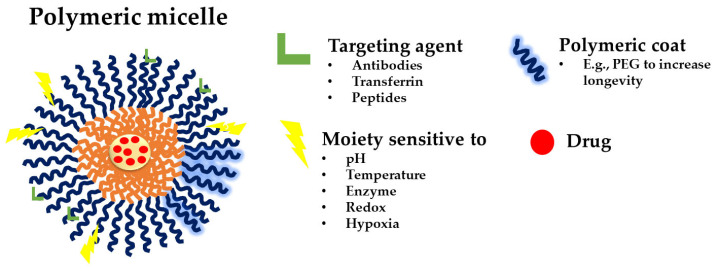
Schematic representation of a polymeric micelle functionalized with different moieties.

**Figure 2 pharmaceutics-15-00976-f002:**
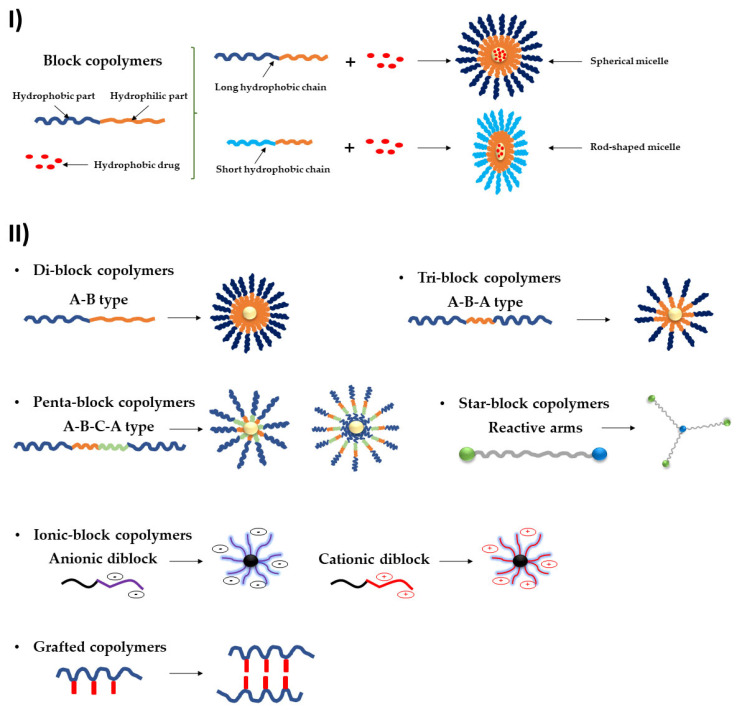
The structure of polymeric micelles. (**I**) Block polymers forming different shaped polymeric micellar systems. (**II**) Example of different block copolymers.

**Figure 3 pharmaceutics-15-00976-f003:**
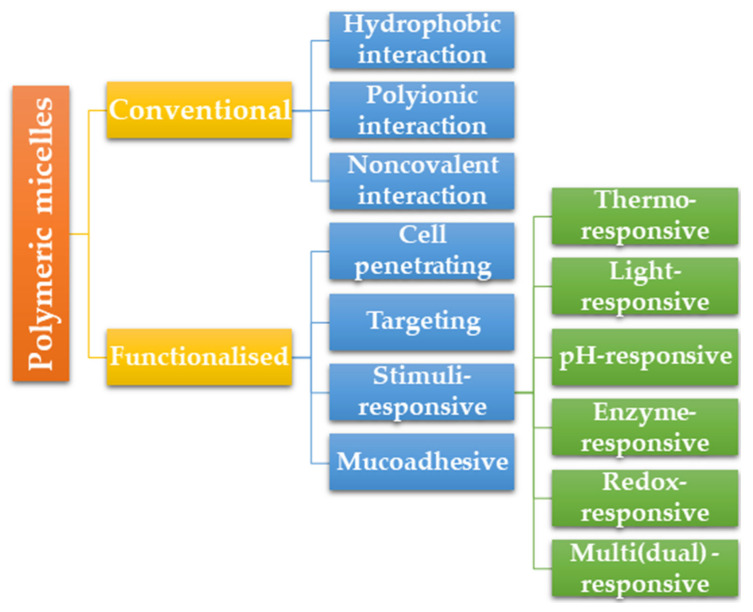
Types of polymeric micelles based on their functions.

**Figure 4 pharmaceutics-15-00976-f004:**
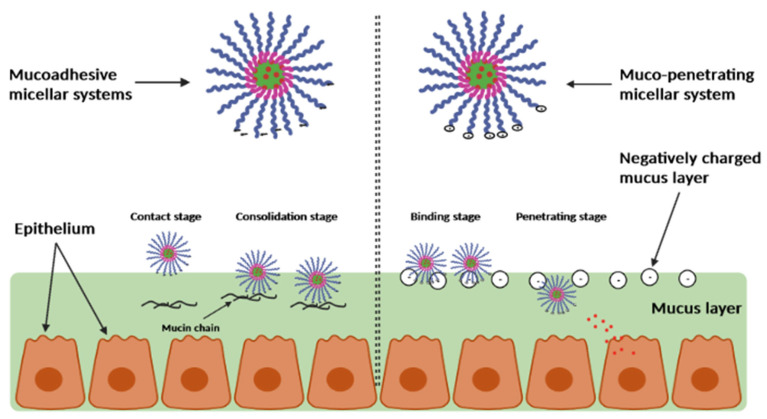
Schematic representation of the mucoadhesive and mucus-penetrating strategy with polymeric micellar systems.

**Figure 5 pharmaceutics-15-00976-f005:**
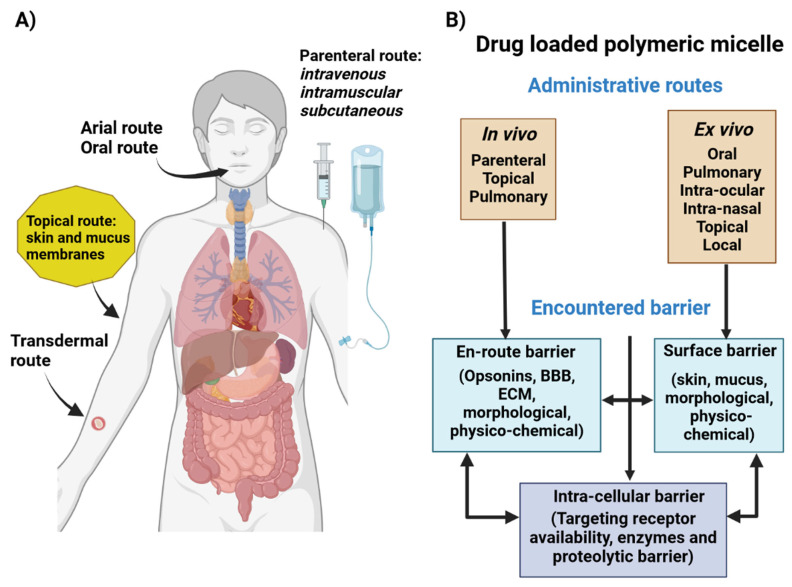
(**A**) Routes of nano-drug administration and (**B**) various barriers encountered by micellar systems before they reach the tumor site.

**Figure 6 pharmaceutics-15-00976-f006:**
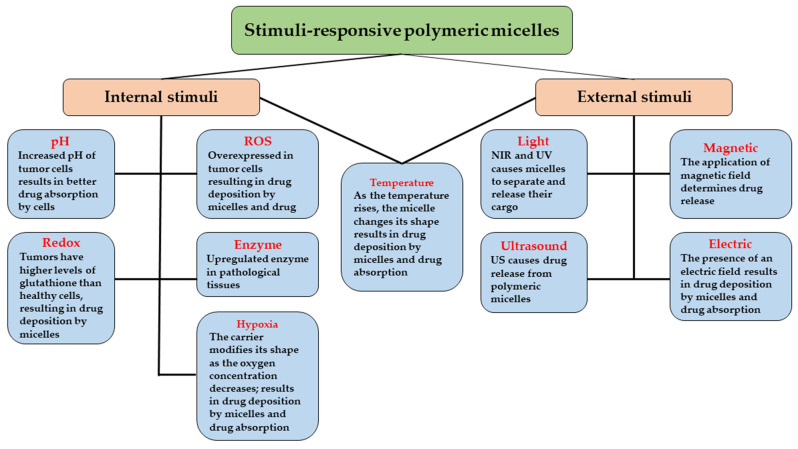
Classification of stimuli-sensitive polymeric micelles.

**Figure 7 pharmaceutics-15-00976-f007:**
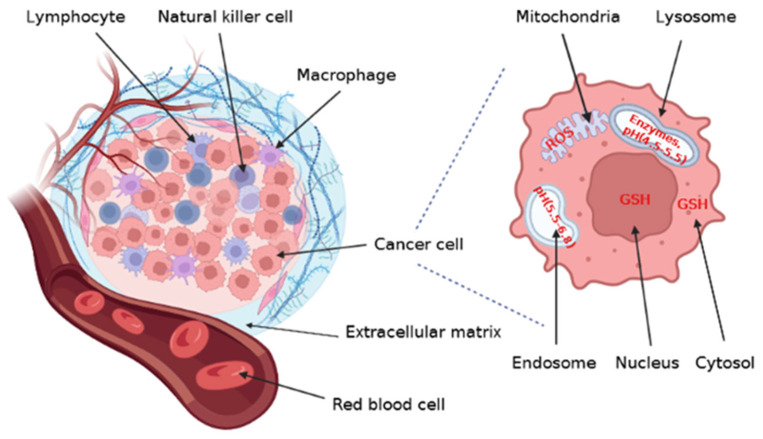
The tumor microenvironment and a cancerous cell—representing various endogenous stimuli.

**Figure 8 pharmaceutics-15-00976-f008:**
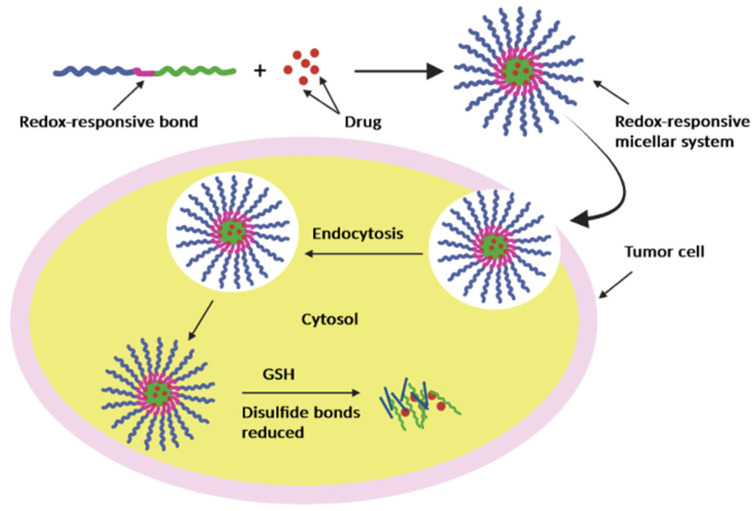
Redox-responsive micelles’ method of action is depicted schematically. The micellar system enters the cancer cell by endocytosis and it actively releases the biologically active chemical into the cytosol as a result of GSH-triggered disintegration.

**Figure 9 pharmaceutics-15-00976-f009:**
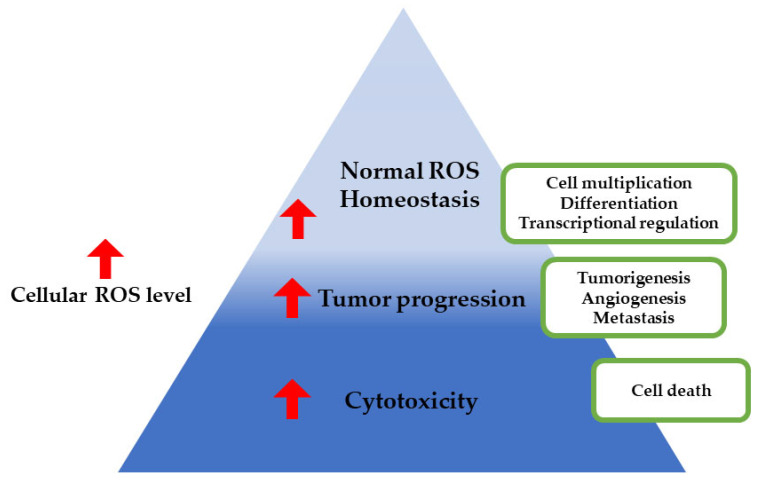
Different ROS concentrations in tumor cells are graphically represented.

**Figure 10 pharmaceutics-15-00976-f010:**
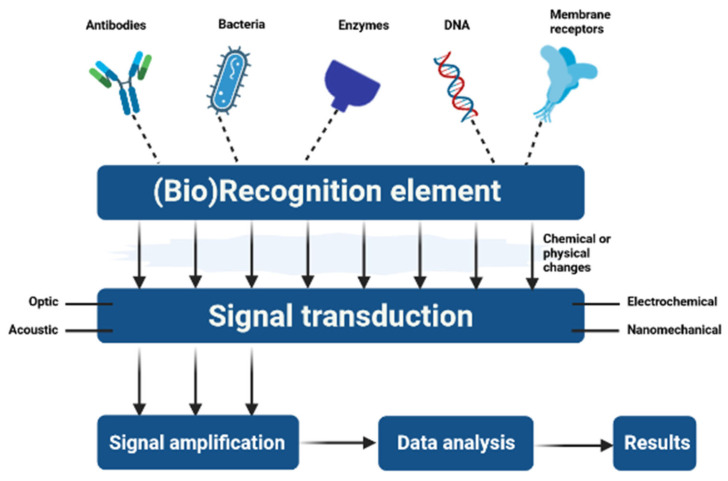
Operational model of a typical biosensor. The biological component functions as a (bio)recognition element, capable of detecting a particular biological element, and is either integrated with or closely related to the physical transducer. Following the interaction, a physical transducer will transform the bio(chemical) signal into a quantifiable discrete or continuous signal, whose intensity may be directly or inversely proportional to the analyte concentration.

**Table 1 pharmaceutics-15-00976-t001:** Examples of hydrophobic and hydrophilic polymers with their corresponding properties, as well as amphiphilic block copolymers frequently utilized in the creation of polymeric micelles.

	Polymer	Chemical Structure	Properties	Ref.
**Hydrophilic polymers**	PEG	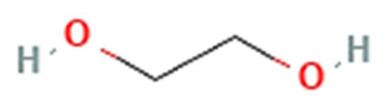	Has been used in clinically approved nanoformulations including polymeric micelles (Genexol^®^ PM).	[12,14,15,16]
	Dextran	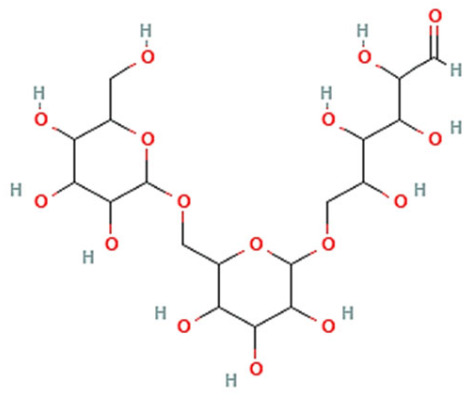	Included into block and graft copolymers as a component. Has been utilized as an excipient in clinically approved injectable pharmaceuticals (Feraheme^®^). Variable molecular weight. Biodegradable.	
	Poly(N-(2-hydroxypropyl)methacrylamide) (PHPMA)	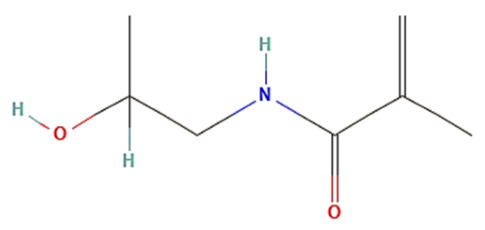	Biocompatible, non-toxic, non-charged, and non-immunogenic.	
**Hydrophobic polymers**	PLA	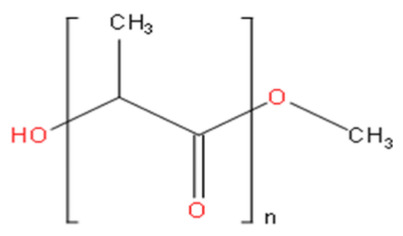	Clinically tested PLA-based polymeric micellar systems (Genexol^®^ and Nanoxel^®^) are available.	
	PLGA	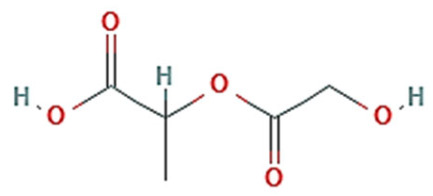	Clinicians utilize PLGA (Vicryl^®^) as a biodegradable surgical suture. Biodegradable.	
	Poly(β-benzyl-l-aspartate)	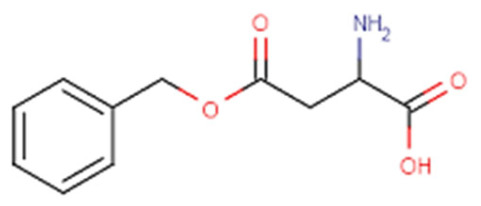	The benzyl group’s presence increases hydrophobicity. Biodegradable.	
	Poly(γ-benzyl-α, l-glutamate)	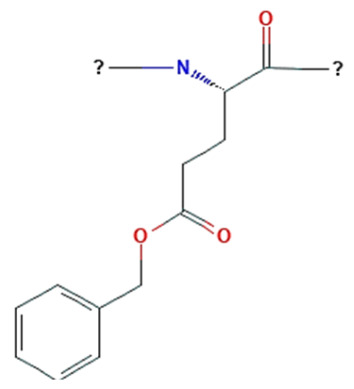	The hydrophobicity can be adjusted by benzyl group’s presence. Extremely high loading capacity for a variety of poorly soluble medications (such as paclitaxel and etoposide). Broad library of polymer architectures.	
**Amphiphilic block copolymers**	Poly(ethylene oxide)–PEO	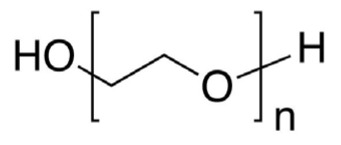	PEO_n_-PPO_m_-PEO_n_ copolymers are frequently employed in pharmaceutical formulations as non-active pharmaceutical components. Clinical trials for SP1049C, Pluronic^®^-based PMs entrapping (DOX). Marketed as poloxamers (Pluronic^®^). Biocompatible.	
	PEO_n_-PPO_m_-PEO_n_			

**Table 2 pharmaceutics-15-00976-t002:** Summary of common methods used for obtaining polymeric micelles.

Method	Advantages	Disadvantages	Ref.
Direct dissolution	- Suitable for extremely hydrophilic molecular low weight polymers - Simple method - Organic approach (without solvents) - Possibility of scaling up	- A minimal medication load - Not appropriate for the majority of hydrophobic copolymers or drugs	[25,26]
Thin-film hydration/solvent evaporation	- High capacity for loading drugs - Possibility of scaling up	- Only usable for copolymers with high hydrophilic–lipophilic balance - Difficulty in removing the free drug and organic solvents from the formulation - Requires expensive equipment - Time-consuming	[25]
Oil in water emulsion	- High capacity for loading drugs - Restricted size range of micelles	- Using chlorinated solvents in the formulation makes it difficult to remove the free drug and organic solvents, which is not optimal for the environment. - Unrealistic for scaling up	[25]
Dialysis	- High capacity for loading drugs - Commonly employed for formulations at the laboratory scale	- Difficulty in eliminating organic solvents and free drugs from the formulation - Unrealistic for scaling up - Time-consuming - Significant amounts of water are necessary	[27]

**Table 3 pharmaceutics-15-00976-t003:** Short summary of various pH-responsive polymeric micellar systems used for the treatment of cancers.

Polymers	Anticancer Agent	Cancer Cell Lines	Experimental Condition & Therapeutic Outcomes	Ref.
Poly(L-lactic acid)-*b*-polylysine/poly(D-lactic acid)-*b*-methoxy poly(ethylene glycol) (PLLA-*b*-PLys/PDLA-*b*-mPEG)	DOX	4T1 breast cancer cells	**Exp. Cond.:** In vitro **Outcomes:** - The DOX-loaded micelles exhibited a slower drug release behavior and a weaker efficacy of intracellular proliferation inhibition than PLLA-b-PLys micelles - DOX-loaded stereocomplexed micelles exhibited lower growth inhibition efficiency of both HepG2 and 4T1 cells	[113]
Poly(ethylene glycol)-imino-poly(benzyl-l-aspartate) (PIPAH)	Hydroxycamptothecin	MCF-7 cells	**Exp. Cond.:** In vitro and in vivo **Outcomes:** - The particle size of the PIPAH micelles did not change at pH 7.4 but increased at pH 6.0 and 5.0, respectively. - Only 25.5% of the encapsulated drug was released under normal physiological conditions (pH 7.4) within 12 h; drug release from the PIPAH micelles in pH 6.0 and 5.0 buffer solutions was noticeably accelerated - Delivery using the PPAH micelles resulted in substantially lower tumor accumulation	[114]
Poly{α-[4-(diethylamino)methyl-1,2,3-triazol]-caprolactone-co-caprolactone}-b-poly(2-methacryloyloxyethylphosphorylcholine) (PDCL-PMPC)	DOX	4T1 breast cancercells	**Exp. Cond.:** In vitro, in vivo, ex vivo **Outcomes:** - The accumulation release account of DOX from PDCL-PMPC micelles was 20% at pH 7.4, but up to 75% at pH 5.0 - At low pH such as a subcellular organelle environment, PDCL-PMPC micelles release DOX quickly - Reasonable in vivo antitumor effect	[115]
Heparin-alpha-tocophero- cis-aconitic anhydride (HEP-CA-TOC)	Docetaxel	MCF-7 and 4TI breast cancercells	**Exp. Cond.:** In vitro **Outcomes:** - Docetaxel release rate increased as the pH of the medium decreased - In the acidic environment of the tumor cells, CA bond starts to hydrolyze and accelerate the release and endosome escape of docetaxel leading to more cytotoxicity and treatment efficiency	[116]
Poly (acrylic acid)-b-polycaprolactone (PAA-b-PCL)	Gambogenic acid	HepG2 cells	**Exp. Cond.:** In vitro, in vivo **Outcomes:** - Only 16.64 % of gambogenic acid was released within the first 12 h - The encapsulation by micelles could enhance both the cytotoxicity and the circulation of gambogenic acid in the body	[117]
Eudragit^®^ S100	Quercetin	CT26 murine colon carcinoma cells	**Exp. Cond.:** In vitro **Outcomes:** - In vitro release testing showed a delay in drug release in acidic pH, but complete release within 24 h at pH 7.2. - Dose-dependent decrease in cell viability	[118]
Cyclodextrins- Acrylic/maleic copolymer	Capecitabine		**Exp. Cond.:** In vitro **Outcomes:** - The release of Capecitabine at pH 1.2 within 2 h is only 4%; at pH 7.4, 70.4% was released within 12 h and 97% was released within 36 h	[119]
N-deacetyl hyaluronic acid dodecylamine	DOX	MCF-7 cells	**Exp. Cond.:** In vitro and in vivo **Outcomes:** - Micelles possessed excellent serum stability at pH 7.4 and very low cytotoxicity - Micelles exhibited antitumor therapeutic efficacy with remarkably low systemic toxicity in vivo	[120]

**Table 4 pharmaceutics-15-00976-t004:** Examples of dual/multiple responsive micellar systems.

Dual/Multiple Responsive	Micellar System	Anti-Cancer Medication	Ref.
**Dual-responsive**			
**pH/Thermo**	Poly (β-amino ester)-grafted disulfide methylene oxide poly (ethylene glycol) (PAE-g-DSMPEG)	DOX	[182]
	PEG-poly (ω- pentadecalactone-co-N-methyldiethyleneamine sebacate-co-2, 2′-thiodiethylene sebacate) (mPEG-b-PAE-ss-DOX, mPEG-b-PAE-cis-DOX)		[183]
	Methoxypoly(ethylene glycol)-cystamine-poly(L-glutamic acid)-imidazole (mPEG-SS-PGA-IM)	PTX	[184]
	PEG-poly(tetrapheny-lethene-co-2-azepane ethyl methacrylate) (mPEG-P(TPE-co-AEMA)	DOX	[185]
	Polyphosphazene (PPZ)	DOX	[186]
	PEG-b-poly(acrylamide-co-acrylonitrile-co-vinylimidazole) copolymer (mPEG-PAAV)	DOX and IR780 (NIR absorber)	[187]
	Poly(methacrylic acid)-b-poly(N-isopropylacrylamid (PMAA-b-PNIPAM)	DOX	[188]
**pH/enzyme**	Hydrophobic modified alginate	DOX	[189]
**Redox/ROS**	PEG(-b-PCL-Ce6)-b-PBEMA		[190]
**ROS/enzyme**	N-isopropylacrylamide (NIPAM)–boronic esters		[191]
**Light/thermo**	Poly(dithienyl-diketopyrrolopyrrole) (PDPP–F127)		[192]
**Light/redox**	PEG- hydrophobic o-nitrobenzyl methacrylate (mPEG-SS-pONBMA)		[193]
**Redox/thermo**	mPEG2k-b-400DTPA-b-mPEG2 (PEG-DTPA)	Nile Red (NR)	[194]
**Multiple-responsive**			
**pH/redox/UV**	PCL -b-poly (acrylic acid) -b-poly (poly (ethylene glycol) methyl ether methacrylate) (6AS-PCL-PAA-PPEGMA)	DOX	[195]
**Redox/temperature/pH**	Poly(γ-benzyl-L-glutamate) (PBLG) and N-isopropylacrylamide (NIPPAM)		[196]
**Temperature/pH/redox/UV**	Poly(ethylene glycol)-ss-[poly(dimethylaminoethyl methacrylate)-copoly(2-nitrobenzyl methacrylate)] [PEG-ss-(PDMAEMA-co-PNBM)]	NR	[197]
**Light/pH/temperature/redox**	Azobenzene-based amphiphilic copolymers P (MMA-co-PEGMA-co-NIPAM-co-HAZOMA) with ionized carboxyl and P (MMA-co-PEGMA-co-NIPAM-co-NNAZOMA)		[198]

**Table 5 pharmaceutics-15-00976-t005:** Polymeric micellar products under market or under different stages of clinical trials.

Trade Name	Drug/ Ingredient	Copolymer	Development Stage	Properties	Identifier	Ref.
Genexol^®^—PM	PTX	mPEG-b-PDLLA	Approved in South Korea, Philippines, India, Vietnam, and Indonesia; Phase I-IV	Improved or equivalent therapeutic efficacy for metastatic breast cancer, non-small cell lung cancer, advanced gastric cancer, and irresectable thymic epithelial tumors; increased drug dosage without premedication and added toxicity	NCT00111904 NCT01023347 NCT00886717 NCT01276548 NCT00882973 NCT00877253	[222]
Nanoxel^®^	PTX	PVP-b-PNIPAM	Approved in India; Phase I-III	Similar therapeutic efficacy, and fewer side effects when compared to Taxol^®^; treating advanced breast cancer, but potential clinical trial deficiency	NCT03614364 NCT04066335 NCT00915369 NCT02639858 NCT03585673 NCT02982395	[222]
Nanoxel^®^ M	Docetaxel	mPEG-b-PDLLA	Approved in South Korea; Phase I-III	Similar PK profiles and antitumor efficacy compared to Taxotere^®^; reduced side effects	NCT04066335 NCT02639858 NCT03585673 NCT02982395 NCT03614364 NCT00915369	[222]
Zisheng^®^	PTX	mPEG-PDLLA	Approved in China; Phase I, III	Enhanced therapeutic efficacy and good patient acceptance for the treatment of non-small cell lung cancer	CTR20150217 CTR20130637 ChiCTR-ONC-14005123 ChiCTR-IPR-15006252	[222]
NK105	PTX	mPEG-b-modified P(Asp)	Phase III	Increased plasma AUC, a decreased occurrence of peripheral sensory neuropathy, and similar overall effectiveness to Taxol ^®^ for metastatic or recurring breast cancer were all observed	NCT01644890	[222]
NK911	DOX	PEG-b-P(Asp-DOX)	Phase I	A better set of PK characteristics when compared to Doxil and the free medication; no fusion-related events; early indications of a therapeutic response	-	[222]
NK012	SN-38	PEG-b-P(Glu-sN-38)	Phase I, II	Patients with unresectable metastatic colorectal cancer experienced similar treatment effectiveness and decreased incidents of serious diarrhea compared to those receiving standard care; patients suffering with refractory lung cancer experienced two complete responses and a 22% overall response rate	NCT00951054 NCT00951613 NCT00542958 NCT01238939 NCT01238952	[222]
NC-6004	Cisplatin	PEG-b-P(Glu)	Phase I-III	Increased tolerability in patients with advanced solid tumors and decreased cisplatin-related toxicity	NCT02043288 NCT02817113	[222]
NC-4016	DACHPt	PEG-b-P(Glu)	Phase I	In a gastric cancer xenograft model, combination treatment with NC-6300 and NC-4016 had superior anticancer efficacy and reduced toxicity	NCT03168035	[222]
K-912/NC-6300	Epirubicin	PEG-b-P(Asp-epirubicin)	Phase I-II	Less toxic than traditional epirubicin, patient tolerance with a variety of solid tumors; early angiosarcoma activity	NCT03168061	[222]
CPC634 (CriPec^®^)	Docetaxel	mPEG-b-P(HPMAm-Lac_n_ -docetaxel)	Phase I-II	Early indications of anti-tumor activity and increased intratumoral docetaxel exposure were noted, as was high dosage skin toxicity	NCT02442531 NCT03742713 NCT03712423	[222]
BIND-014	Docetaxel	PLA-PEG-GL	Phase I-II	Antitumor activity in metastatic castration-resistant prostate cancer; good tolerance and tolerable toxicity; clinically active and well-tolerated in stage III/IV non-small cell lung cancer	NCT01812746 NCT01792479 NCT01300533 NCT02283320 NCT02479178	[222]
SP1049C	DOX	Pluronic^®^ L61/F127	Phase I-II	Patients suffering from gastroesophageal junction and esophageal cancer showed therapeutic effectiveness as a single drug and a tolerable safety profile	-	[222]
Resveratrol from Vineatrol 30^®^ extract incorporated into micelles	Resveratrol		Phase I	Estimation of pharmacokinetics and safety	NCT02944097	[222]
PTX micelles	PTX		Phase I	Investigated for efficacy in advanced solid tumors	NCT04778839	[222]
PTX micelles for injection	PTX		Phase I	Investigated for efficvacy in gynecological cancer	NCT02739529	[222]
ONM-100	Indocyanine green		Phase II	Investigated for the detection of cancer in patients with solid tumors (breast cancer, head and neck squamous cell carcinoma, colorectal cancer, prostate cancer, ovarian cancer, urothelial carcinoma, non-small cell lung cancer) undergoing routine surgery	NCT03735680	[222]

**Table 6 pharmaceutics-15-00976-t006:** Options for overcoming the limits of polymeric micelles usage.

Limitations	Options	Ref.
**Low drug loading**	- increasing the drug-polymer compatibility - cross-linking of the core and shell of self-assembled polymeric micelles - electrostatic interactions - the micelles can be coated one layer at a time - particles that resemble micelles - lipids should incorporate drug-attached polymers - polymeric prodrugs	[228,229]
**High CMC**	- lengthening the hydrophobic block’s chain - micelle cores can be decorated with different fatty acids, and a benzyl group can be added.	[230]
**Rapid clearance**	- PEGylation approach - Cross-linking with stimuli-sensitive linkers	[231]
**Low efficiency in drug delivery**	- the use of high-affinity targeting ligands - cross-linking with diverse stimuli-sensitive linkers - intracellular redox-responsive drug release	[232,233]
**Low selectivity**	- PEGylation approach - the use of high-affinity targeting ligands	[228,229,231]
**Low capacity to disrupt membranes**	- the use of hydrophobic moieties and cationic groups - the use of polymers with buffering capacity at endosomal pH - the use of high-affinity targeting ligands	[234,235]
**Low stability**	- PEGylation approach - Covalent cross-linking methods that result in shell cross-linked micelles, or core cross-linked ones - covalent cross-linking strategies: photo/UV dimerization, di-functional cross-linkers, click cross-linking method, silicon chemistry method, and reversible boronate ester bond - non-covalent cross-linking by means of micelle cores complexation - modifying the hydrophilic/hydrophobic block ratios of the micelles - increasing the crystallinity of hydrophobic groups - inorganic materials introduction into the core or shell in order to function as structural stabilizers	[230,236]
**Toxicity and immunogenicity**	- PEGylation approach - using pH-sensitive micelles - using high affinity targeting ligands - using biodegradable and biocompatible micellar systems	[142,237]
**Non-biodegradability and non-biocompatibility**	- the use of biodegradable micellar systems such as: PEG, PLA, PCL, mPEG-PDLLA, poly(L-histidine)	[238]

## Data Availability

Not applicable.

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
