# Peer review of "Polymeric Micellar Systems—A Special Emphasis on “Smart” Drug Delivery"

_pharmaceutics, 2023, doi:10.3390/pharmaceutics15030976_

Round 1

Reviewer 1 Report

For Table 1, classifying polymers as hydrophobic and hydrophilic is not correct, especially assuming that most of them are still amphiphilic. The properties provided in this Table vary based on basic polymer characteristics such as molecular weight and PDI, so the information in the  Table looks vague to me. Besides, both hydrophobicity and hydrophilicity are relative characteristics. When Authors say “hydrophobicity is increased”, there is always a question “In comparison to what”. Overall, I recommend Authors would rethink the Table 1 content and layout. Also, it is always good to see chemical structures and compositions, not just the names.

Section “Preparation methods” does not contain new information to be reviewed. All information there has been included in multiple reviews published several decades ago. I would eliminate this part from the review including Figure 1 and Table 2

By all respect, Authors disregarded reviewing a large fraction of presented polymeric systems in terms of their stimuli-responses. In particular, micellar polymer candidates sensitive to environmental polarity changes (in particular, when polymers self-assembled carrier migrates from an aqueous phase and delivers cargo to biomembrane surface) are not covered.

Poor quality of some figures, including Figure 2. Besides, this Figure shows micelles from diblock and triblock copolymers. Are those only known macromolecular configurations yielding the micellar structures for drug delivery? I struggle with nomenclature for micelles used in Figure 2

I recommend revising the title into the more academic one. “Intelligent” is still a slang word.

I recommend major revision.

Author Response

  1. For Table 1, classifying polymers as hydrophobic and hydrophilic is not correct, especially assuming that most of them are still amphiphilic. The properties provided in this Table vary based on basic polymer characteristics such as molecular weight and PDI, so the information in the Table looks vague to me. Besides, both hydrophobicity and hydrophilicity are relative characteristics. When Authors say “hydrophobicity is increased”, there is always a question “In comparison to what”. Overall, I recommend Authors would rethink the Table 1 content and layout. Also, it is always good to see chemical structures and compositions, not just the names.

R1. Thank this useful suggestion and comments. We agree with the reviewer that “hydrophobicity is increased in comparison to what”. Therefore, we used more appropriate wording. We did our best to make a more specific table, in which we also included the chemical structures of the polymers. Moreover, we revised the text regarding Table 1, and we did some modifications. 

  1. Section “Preparation methods” does not contain new information to be reviewed. All information there has been included in multiple reviews published several decades ago. I would eliminate this part from the review including Figure 1 and Table 2.

R2. We are thankful to this reviewer for this suggestion. Indeed, this part of the review can be found in many papers. However, we kept in mind that these techniques have been presented in other reviews or can be the subject of another substantial review. Thus, to raise awareness and to introduce the reader to the “world” of world” of micelles preparation methods, we shortly summarised the methods. However, we took the liberty of keeping this part in our review and removing only Figure 1.

  1. 3. By all respect, Authors disregarded reviewing a large fraction of presented polymeric systems in terms of their stimuli-responses. In particular, micellar polymer candidates sensitive to environmental polarity changes (in particular, when polymers self-assembled carrier migrates from an aqueous phase and delivers cargo to biomembrane surface) are not covered.

R3. Thank you for suggesting this topic of drug delivery by polymeric micelles. We are aware that micellar polymer candidates can be “destabilized” by changing polymers polarity: The authors did not disregard this topic as it also represents an attractive tool for drug delivery; we only talked about drug loading taking into account the polarity of the medication (2.1. Polymeric micelles and micellar structures). Moreover, we consider that the polarity changes of different biological membranes and, paralleled, the micelles sensitive to these changes can be future work on micellar structures for drug delivery.

  1. 4. Poor quality of some figures, including Figure 2. Besides, this Figure shows micelles from diblock and triblock copolymers. Are those only known macromolecular configurations yielding the micellar structures for drug delivery? I struggle with nomenclature for micelles used in Figure 2

R4. We agree with Reviewer 1 regarding all aspects of Figure 2, and we are thankful for pointing out the deficiencies. Therefore, based on this comment, we re-created Figures 2 and z. We introduced other macromolecular configurations in Figure 2 and increased their quality. We hope that now it will add more value to our manuscript. Moreover, we added more information regarding the configurations of the micellar systems concerning the new Figure 2 and the Reviewer’s remarks.

  1. 5. I recommend revising the title into the more academic one. “Intelligent” is still a slang word.

R5. Considering your recommendation and Reviewer 4 suggestion, we used “smart” instead.

I recommend major revision.

Reviewer 2 Report

Comments: The review describes a very interesting topic in advanced nanotechnology today providing many information and description. However many critical points should clarify

1-    Despite the importance of this study, however, critical micelle concentration CMC) was completely missed.  A new title should be written with CMC, including a deep description for this specific name. Also, the authors did not provide criteria for how to choose suitable polymers for producing optimal micelles.

2-    Chemical scheme to illustrate how micelles could be optimized,  is mostly missed. Authors should provide simple desecration for the reader by doing a chemical scheme.

3-    The mechanism by which micelles can be functionalized is still also not described well.  Authors should provide in detail information on how micelles' surfaces could be functionalized.

4-    In the “Mucoadhesive and mucus-penetrating polymeric micelles section” the authors didnot explain how micelles can penetrate mucous layers. please provide a biological scheme.

5-    In the " Redox" section would you please draw a chemical scheme to provide a more clear description for readers?

6-    Refs should be arranged according to a style of pharmaceutics MDPI Journal.

7-    In recent advanced pharmaceutical technology, authors should include their own opinion and their biomedical suggestions on how to improve the outcome of micelles

8-    The manuscript lacks a comprehensive study to illustrate why there is still a worse outcome in the treatment of cancer and how encapsulated chemotherapies in cancer could make a difference.

9-    How micelles can penetrate adherent junctions was not studied.  

Author Response

Comments: The review describes a very interesting topic in advanced nanotechnology today providing many information and description. However many critical points should clarify

1-   Despite the importance of this study, however, critical micelle concentration CMC) was missed entirely.  A new title should be written with CMC, including a deep description of this specific name. Also, the authors did not provide criteria for how to choose suitable polymers for producing optimal micelles.

R1. Thank you for this suggestion. We added a subchapter dedicated to CMC, “2.2. Critical micelle concentration”, and a description of this parameter. We hope that this will add value to our work.

2-   Chemical scheme to illustrate how micelles could be optimized, is mostly missed. Authors should provide simple desecration for the reader by doing a chemical scheme.

R2. We introduced a new chapter, “6. Advantages and drawbacks of polymeric micellar systems” in which we tabulated the drawbacks of polymeric micelles, and provided, at the same time, some options for optimising these systems.

3-    The mechanism by which micelles can be functionalized is still also not described well.  Authors should provide in detail information on how micelles' surfaces could be functionalized.

R3. We introduced a new subchapter named “2.3.6. Functionalization methods”, in which we present information on how the surfaces of micelles can be functionalised.

4-   In the “Mucoadhesive and mucus-penetrating polymeric micelles section” the authors did not explain how micelles can penetrate mucous layers. please provide a biological scheme.

R4. We provided a scheme regarding the mucoadhesive and mucus-penetrating polymeric micelles. We hope that this scheme will provide some insights into this particular type of system.

5-   In the "Redox" section would you please draw a chemical scheme to provide a more clear description for readers?

R5. Thank you for your suggestion. We added a scheme in the “Redox” Section.

6-    Refs should be arranged according to a style of pharmaceutics MDPI Journal.

R6. Thank you for this suggestion. We use a program for references that already arranged them according to MDPI Journal.

7-   In recent advanced pharmaceutical technology, authors should include their own opinion and their biomedical suggestions on how to improve the outcome of micelles

R7. Phrases regarding our opinion were already introduced in the Conclusions section.

8-   The manuscript lacks a comprehensive study to illustrate why there is still a worse outcome in the treatment of cancer and how encapsulated chemotherapies in cancer could make a difference

R8. As we mentioned above, we made a new chapter, “6. Advantages and drawbacks of polymeric micellar systems”, in which we tabulated the drawbacks of polymeric micelles and we provided, at the same time, some options for optimising these systems.

9-   How micelles can penetrate adherent junctions was not studied. 

R9. Thank you for this suggestion. Based on this, we introduced a new subchapter, “2.6. Biological barriers and polymeric micelles for efficient anticancer therapeutic drug delivery”, dedicated to the biological barriers and polymeric micelles crossing. Moreover, the Reviewer can find a scheme for mucoadhesive and mucus-penetrating polymeric micelles. We hope that both additions are of high value to our manuscript.

Reviewer 3 Report

The review on “Polymeric micellar systems - A special emphasis on “intelligent” drug delivery” is an in-depth review focusing on summarizing the state-of-the-art and the most recent developments of polymeric micellar systems with respect to cancer treatments.

Suggestions and comments for improvement of the manuscript are as follows:

1. Please add an Image of micelles in the introduction. 

2. Improve the resolution of figure 2.

3. Figure 5. The tumor microenvironment and a cancerous cell - represent various endogenous stimuli, the caption should be below the figure. 

4. Like  Table 3. Short summary of various pH- responsive polymeric micellar systems used for the treatment of cancers, all micellar systems should be summarized in all in one table.

5. Line 178-719: Normal ROS levels promote the growth of malignant cells. Tumor metastasis results from moderate levels. ROS that are toxic cause cell death. Not get the context. 

Author Response

The review on “Polymeric micellar systems - A special emphasis on “intelligent” drug delivery” is an in-depth review focusing on summarizing the state-of-the-art and the most recent developments of polymeric micellar systems with respect to cancer treatments.

Suggestions and comments for improvement of the manuscript are as follows:

  1. Please add an Image of micelles in the introduction. 

R1. We added an image of the micelles in the Introduction section.

  1. 2. Improve the resolution of figure 2.

R2. Thank you for this remark. Also, we took into account the remarks of Reviewer 1 on the same image, and we made a new Figure 2 with an improved resolution.

  1. Figure 5. The tumor microenvironment and a cancerous cell - represent various endogenous stimuli, the caption should be below the figure. 

R3. Thank you for highlighting this inadequacy to us. We put the caption where it was supposed to be.

  1. 4. Like Table 3. Short summary of various pH- responsive polymeric micellar systems used for the treatment of cancers, all micellar systems should be summarized in all in one table.

R4. Table 3 is meant as a summary. The scientific literature on the pH-responsive micellar systems used for cancer treatment is enormous. In a basic search on WOS, in the last five years, there are more than >800 scientific articles on “pH-responsive micelles for cancer treatment”. Therefore, considering that our paper is not dedicated only to pH-responsive micellar systems, we only selected a few articles to be introduced in the table.

  1. 5. Line 178-719: Normal ROS levels promote the growth of malignant cells. Tumor metastasis results from moderate levels. ROS that are toxic cause cell death. Not get the context. 

R5. We took the liberty and deleted the text which was out of context.

Reviewer 4 Report

1. amphiphilic block copolymers (ABSs). ABCs?

2. In Table 1, for Poly(2-n-butyl-2-oxazoline), the authors mentioned that the benzyl group's presence increases hydrophobicity. Poly(2-n-butyl-2-oxazoline) does not have benzyl group.

3. What's the advantages and drawbacks of polymeric micelle system?

4. The authors should discuss the barriers for drug delivery and how the polymeric micelle can overcome these barriers through advancing the functionality of nanomedicine. The recent references from Kataoka group (a pioneer of polymeric micelle delivery systems) may be useful for this review (https://doi.org/10.1021/jacs.0c09029; https://doi.org/10.1016/j.jconrel.2022.03.049). Based on these, the authors also should give some strategies for next-generation micelle delivery system.

5. I do not suggest the authors use the 'intelligent'. In fact, current delivery system is far from 'intelligent system'. Please use 'smart' instead.

6. Drug release is important final step of drug delivery. An important biological hypothesis for nanomedicine is that the diseased tissue microenvironment can trigger a desirable event to a large extent, such as drug release derived from stimuli-responsive behavior. However, this is not always the case, based on our rudimentary understanding of microenvironment and nano− bio interaction. In our opinion, toxic drug release/production is a critical determinant of efficacy, unfortunately thus far being poorly evaluated or usually uncharacterized in vivo although a perfect release profile is exhibited in vitro. Recently, the researcher proposed the tissue microenvironment-reprogramming strategies for amplifying responsiveness (https://doi.org/10.1016/j.jconrel.2016.01.029). The authors can discuss on this point.

7. Some young researchers have done a lot of smart micelle delivery system (pH, ROS, enzyme) to increase the function (e.g., tumor penetration, drug release, cellular uptake, and endosomal escape), for example Prof. Zhishen Ge and Prof. Zhen Gu. It is better to include some of them.

Author Response

  1. 1. amphiphilic block copolymers (ABSs). ABCs?

R1. Thank you for pointing out this typo. We modified the word.

  1. 2. In Table 1, for Poly(2-n-butyl-2-oxazoline), the authors mentioned that the benzyl group's presence increases hydrophobicity. Poly(2-n-butyl-2-oxazoline) does not have a benzyl group.

R2. Thank you for pointing out this fact, which we missed. Taking into account your and Reviewer 1 comment, we redesigned Table 1, and we hope that, in this form, it will be suited for our review.

  1. 3. What's the advantages and drawbacks of polymeric micelle system?

R3. Thank you for this question. We introduced in our review a new chapter, named” Advantages and drawbacks of polymeric micelle system”, to better fit the mention of Reviewer 4.

  1. 4. The authors should discuss the barriers for drug delivery and how the polymeric micelle can overcome these barriers through advancing the functionality of nanomedicine. The recent references from Kataoka group (a pioneer of polymeric micelle delivery systems) may be useful for this review (https://doi.org/10.1021/jacs.0c09029; https://doi.org/10.1016/j.jconrel.2022.03.049). Based on these, the authors also should give some strategies for next-generation micelle delivery system.

R4. Thank you for suggesting the introduction of a new discussion. Considering your valuable suggestion, we introduced a new subchapter, “2.5. Biological barriers and polymeric micelles for efficient anticancer therapeutic drug delivery”, dedicated to the biological barriers and polymeric micelles crossing. Moreover, we considered and cited the interesting works of the Kataoka group.

  1. 5. I do not suggest the authors use the 'intelligent'. In fact, current delivery system is far from 'intelligent system'. Please use 'smart' instead.

R5. Thank you for your suggestion. We modified the title accordingly.

  1. Drug release is important final step of drug delivery. An important biological hypothesis for nanomedicine is that the diseased tissue microenvironment can trigger a desirable event to a large extent, such as drug release derived from stimuli-responsive behavior. However, this is not always the case, based on our rudimentary understanding of microenvironment and nano− bio interaction. In our opinion, toxic drug release/production is a critical determinant of efficacy, unfortunately thus far being poorly evaluated or usually uncharacterized in vivo although a perfect release profile is exhibited in vitro. Recently, the researcher proposed the tissue microenvironment-reprogramming strategies for amplifying responsiveness (https://doi.org/10.1016/j.jconrel.2016.01.029). The authors can discuss on this point.

R6. Thank you for this comment. We discussed the proposed strategy in the subchapter “3.1.6. ROS”.

  1. a. Some young researchers have done a lot of smart micelle delivery system (pH, ROS, enzyme) to increase the function (e.g., tumor penetration, drug release, cellular uptake, and endosomal escape), for example Zhishen Ge and Prof. Zhen Gu. It is better to include some of them.

R.a. Thank you for suggesting the exciting works. We included some of their works in our review. The paragraphs dedicated to their research are in “3.1.3. Redox”.

Round 2

Reviewer 1 Report

Manuscript has been improved significantly. Speaking about "intelligent delivery", the current review version is still missing information and references on widely reported in literature amphiphilic invertible polymers (multiblock copolymers) responsive to environmental polarity. Those polymers undergo conformational changes at biomembrane, and, thus, become promising polymeric materials for drug delivery. This could be helpful information to add.

Author Response

Reviewer 1.

Manuscript has been improved significantly. Speaking about "intelligent delivery", the current review version is still missing information and references on widely reported in literature amphiphilic invertible polymers (multiblock copolymers) responsive to environmental polarity. Those polymers undergo conformational changes at biomembrane, and, thus, become promising polymeric materials for drug delivery. This could be helpful information to add.

  1. R. Thank you for helping us improve the manuscript. We included information and references on micelles formed from amphiphilic invertible polymers responsive to environmental polarity. This part is included in Chapter 2 2. General issues and status quo of polymeric micelles. Moreover, we dedicated a new subchapter, “3.1.7. Stimuli-responsive inversion of macromolecules”, to micellar systems responsive to changes in environmental polarity.

Reviewer 2 Report

The manuscript was revised point by point according to the reviewer's comments. Manuscript is more acceptable now. 

Author Response

Reviewer 2.

The manuscript was revised point by point according to the reviewer's comments. The manuscript is more acceptable now. 

  1. R. Thank you for your comments. We reread the whole article and made corrections regarding grammar and spelling. The modifications are marked in the text by track changes.

Reviewer 4 Report

The authors addressed the most concerns well. One minor comment: polymeric micellar systems with "smart" responsibility. 'Responsibility' should be 'responsiveness'. Please check the language thoroughly.

Author Response

Reviewer 4.

The authors addressed the most concerns well. One minor comment: polymeric micellar systems with "smart" responsibility. 'Responsibility' should be 'responsiveness'. Please check the language thoroughly.

  1. R. Thank you for pointing out this typo. We corrected it accordingly. Also, we made corrections regarding grammar and spelling.